# Engineering chirality at wafer scale with ordered carbon nanotube architectures

Jacques Doumani[1,2,3], Minhan Lou [3], Oliver Dewey [4,5], Nina Hong[6], Jichao Fan[3], Andrey Baydin [1,7], Keshav Zahn [1], Yohei Yomogida [8], Kazuhiro Yanagi[8], Matteo Pasquali [4,5,7,9,10], Riichiro Saito [8,11,12], Junichiro Kono [1,4,7,10,13] & Weilu Gao [3,4] ✉

Creating artificial matter with controllable chirality in a simple and scalable manner brings new opportunities to diverse areas. Here we show two such methods based on controlled vacuum filtration - twist stacking and mechanical rotation - for fabricating wafer-scale chiral architectures of ordered carbon nanotubes (CNTs) with tunable and large circular dichroism (CD). By controlling the stacking angle and handedness in the twist-stacking approach, we maximize the CD response and achieve a high deep-ultraviolet ellipticity of $40 \pm 1$ mdeg nm$^{-1}$. Our theoretical simulations using the transfer matrix method reproduce the experimentally observed CD spectra and further predict that an optimized film of twist-stacked CNTs can exhibit an ellipticity as high as 150 mdeg nm$^{-1}$, corresponding to a $g$ factor of 0.22. Furthermore, the mechanical rotation method not only accelerates the fabrication of twisted structures but also produces both chiralities simultaneously in a single sample, in a single run, and in a controllable manner. The created wafer-scale objects represent an alternative type of synthetic chiral matter consisting of ordered quantum wires whose macroscopic properties are governed by nanoscopic electronic signatures and can be used to explore chiral phenomena and develop chiral photonic and optoelectronic devices.

Chirality is a degree of freedom in objects with broken mirror symmetry, which ubiquitously exists in both natural and synthetic matter, ranging from molecules through crystals to metamaterials. There is much current interest in studying chiral effects in electronic and photonic processes in materials that can lead to new device concepts and implementations[1–3]. In particular, the interaction of chiral matter with circularly polarized light has profound implications and consequences in sensing[4], plasmonics[5], cryptographic imaging and communication[6–8], and quantum optics[3,9,10].

The key enabler of these diverse applications is a versatile macroscopic chiral platform with large and controllable chiroptical properties. Natural molecules generally have weak chiroptical responses because of their small sizes compared to the light wavelength. Chiral metamaterials, consisting of periodic artificial symmetry-breaking

[1]Department of Electrical and Computer Engineering, Rice University, Houston, TX, USA. [2]Applied Physics Graduate Program, Smalley–Curl Institute, Rice University, Houston, TX, USA. [3]Department of Electrical and Computer Engineering, The University of Utah, Salt Lake City, UT, USA. [4]Carbon Hub, Rice University, Houston, TX, USA. [5]Department of Chemical and Biomolecular Engineering, Rice University, Houston, TX, USA. [6]J.A. Woollam Co., Inc., Lincoln, NE, USA. [7]Smalley–Curl Institute, Rice University, Houston, TX, USA. [8]Department of Physics, Tokyo Metropolitan University, Tokyo, Japan. [9]Department of Chemistry, Rice University, Houston, TX, USA. [10]Department of Materials Science and NanoEngineering, Rice University, Houston, TX, USA. [11]Department of Physics, Tohoku University, Sendai, Japan. [12]Department of Physics, National Taiwan Normal University, Taipei, Taiwan. [13]Department of Physics and Astronomy, Rice University, Houston, TX, USA. ✉e-mail: weilu.gao@utah.edu

structures such as twisted structures manufactured with either top-down or bottom-up methods, can boost chiroptical responses through resonance enhancement[11,12]. However, top-down manufacturing requires sophisticated nanofabrication facilities and intricate processes to create engineered structures[13], especially for short-wavelength applications that necessitate ultrasmall feature sizes. Bottom-up assembling of one-dimensional (1D) objects offers a simple and scalable manufacturing route, while current demonstrations are constrained to simple metallic or dielectric rods with limited conventional material physical properties[14–18]. Nanomaterials with room-temperature quantum-confinement effects and their artificial architectures have recently emerged as chiral platforms. For example, chirally stacked multiple layers of graphene have displayed circular dichroism (CD), which is defined as the differential attenuation of left and right circularly polarized light[19]. Methods that can create artificial matter based on such nanomaterials with strong and controllable chirality without involving complicated procedures are being sought.

Here, we present two simple and scalable procedures, both based on the controlled vacuum filtration (CVF) method[20,21], for preparing wafer-scale chiral matter consisting of ordered carbon nanotubes (CNTs). The first procedure uses twist-stacking of aligned CNT films produced by CVF, whereas the second involves mechanical rotation during CVF. In both approaches, we employed a racemic mixture of CNTs, and hence, the chirality manifested by the produced architectures is not due to the intrinsic chirality of chiral CNTs. CD spectra exhibited spectral peaks reflecting the electronic structure of the underlying individual CNTs, which are quantum wires with 1D van Hove singularities. Through precise control of stacking angle and handedness in the twist-stacking approach, we obtained a high ellipticity of $40 \pm 1$ mdeg nm$^{-1}$ in the deep-ultraviolet (DUV) range (4.43–6.2 eV). We performed electromagnetic simulations based on the transfer matrix method, which reproduced all observed CD and attenuation spectra while simultaneously determining the dielectric functions of the aligned CNT films. We further predict that a film of twist-stacked CNTs with an optimized thickness will exhibit an ellipticity as high as 150 mdeg nm$^{-1}$, which corresponds to a $g$ factor of 0.22. Furthermore, the mechanical rotation method not only accelerated the fabrication of twisted structures but also produced both chiralities simultaneously in a single sample, in a single run, and in a controllable manner. This CNT-based chiral platform will provide opportunities not only for photonic and optoelectronic devices, such as chiral (quantum)

optical emitters[22], chiral optical sensors, and chiral photodetectors but also for exploration of phenomena, including non-optical effects such as spintronics[23], for broader applications.

## Results

### Aligned CNT films produced by CVF

Figure 1a schematically shows the vacuum filtration system we used for CVF, where individually suspended CNTs in water spontaneously assemble to form a wafer-scale, highly aligned, and densely packed film on the filter membrane in a well-controlled manner[20,21,24]; see Methods for more details. As CNTs accumulate near the surface of the filter membrane during the filtration process, a liquid-crystal phase transition leads to wafer-scale alignment of CNTs[20,21]. The CVF process is spontaneous and does not require any external stimulus. Figure 1b shows a photograph, a scanning electron microscopy image, and an atomic force microscopy image of a typically obtained aligned CNT film. The original aqueous suspension contained a racemic mixture of CNTs of both semiconducting and metallic types, showing zero CD (see Supplementary Fig. 1). The experimentally measured quantity of CD is the ellipticity ($\psi$), which is defined as $\psi \equiv \arctan\{(E_l - E_r)/(E_l + E_r)\}$, expressed in units of mdeg. Here, $E_l$ and $E_r$ are the magnitudes of the left- and right-circular components, respectively, of the output electric field when the input beam that enters the material is linearly polarized[25]. We further normalized the ellipticity by the film thickness for the purpose of comparing different samples, so $\psi$ is given in units of mdeg nm$^{-1}$. Figure 1c shows attenuation spectra for an aligned CNT film on a fused silica substrate when the incident light polarization is parallel and perpendicular to the CNT alignment direction. The 1D nature of CNTs leads to strongly anisotropic optical absorption[26]. When the polarization of incident light is parallel to the nanotube axis, strong absorption occurs through transitions between subbands with the same index such as the $S_{22}$ transition in semiconducting CNTs at 1.2 eV, the $M_{11}$ transition in metallic CNTs at 1.7 eV, and the DUV response (labeled '$\pi$') at 4.4 eV. Specifically, the DUV spectral features originate from interband transitions associated with the $M$ point of the graphene Brillouin zone[27,28] and the transition energies depend on the diameters and atomic structures of CNTs[27,29,30]. In contrast, for perpendicular polarization, different interband transitions are excited due to optical selection rules, but their intensities are suppressed because of the depolarization effect[31,32]. The DUV attenuation peak shifts to 4.87 eV for perpendicular polarization. Such a shift is due to the excitation of different interband transitions,

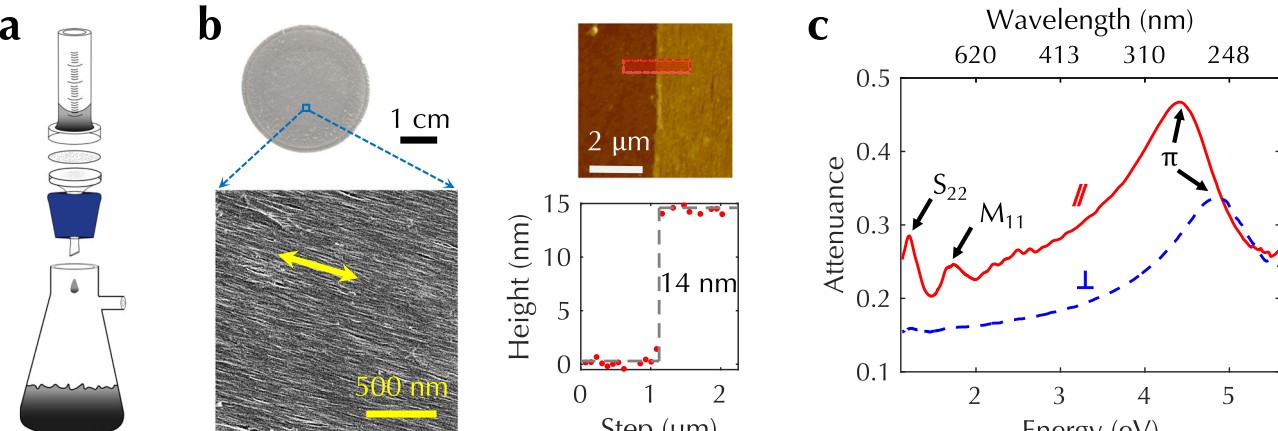

**Fig. 1 | Aligned CNT films produced by CVF and their strongly polarization-dependent optical response. a** Schematic of the vacuum filtration system. **b** Representative photograph (top left), scanning electron microscopy image (bottom left), atomic force image (top right), and height profile (bottom right) of an obtained aligned CNT film. The yellow arrow in the scanning electron microscopy image indicates the CNT alignment direction. The red rectangle in the atomic force

image indicates the height profile measurement region. The red dots in the height profile are experimental data and the gray dashed line is for eye guidance. **c** Polarization-dependent optical attenuation spectra for an aligned CNT film from the near-infrared to the DUV. The red solid line is for parallel polarization and the blue dashed line is for perpendicular polarization. Source data are provided as a Source Data file.

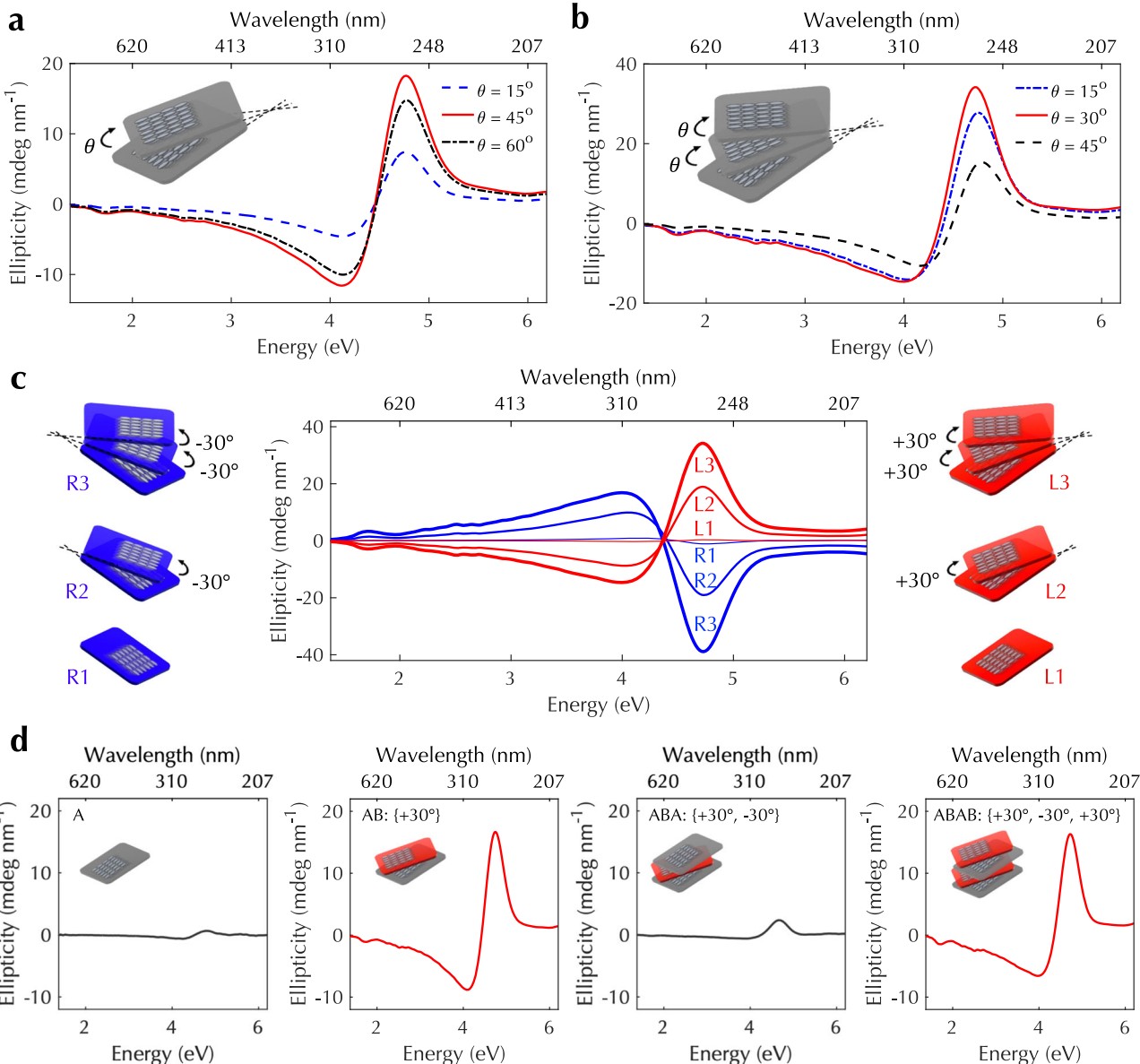

**Fig. 2 | Controlled chirality in twist-stacking-produced multiple-layered architectures of aligned CNTs. a** CD spectra for a twisted two-layer CNT stack at twist angles of 15° (blue dashed line), 45° (red solid line), and 60° (black dashed line). **b** CD spectra for a twisted three-layer CNT stack with two equal twist angles, $\theta = 15°$ (blue dashed line), 30° (red solid line), and 45° (black dashed line). **c** CD spectra for left- (red lines) and right-handed (blue lines) twisted three-layer stacks. All twist angles in left-handed stacks are expressed as positive angles (+30°), while those in right-handed stacks are expressed as negative angles (−30°). **d** CD spectra for CNT architectures in the A, AB, ABA, and ABAB twist configurations. All twist angles have the same absolute value of 30°. Source data are provided as a Source Data file.

while unambiguous assignments of these transitions remain elusive and require systematic studies of aligned CNT films of various diameters and atomic structures[33].

## Twist stacking of aligned CNT films

The first method we developed for creating wafer-scale chiral CNT architectures is based on the twist-stacking of multiple aligned CNT films prepared via CVF. Figure 2a illustrates a twisted two-layer stack. After first transferring an aligned CNT film onto a substrate, we then transferred the other aligned film on top of the first film with a twist angle $\theta$ between two alignment directions. The largest ellipticity occurs when the twist angle is 45°. We then repeated the same procedure for placing a third film on the second film with the same twist angle $\theta$ (Fig. 2b); see Methods and Supplementary Fig. 2 for more details. In the twisted three-layer stack, the largest ellipticity occurs when the angle is 30°. Compared to other methods of creating twisted structures using

huge electric voltage and polymer/CNT composites[34], our method is simpler and more scalable and can produce CNT-only structures.

The observed CD spectra were measured using a standard CD spectrometer in an energy range of 1.38–6.2 eV with a customized 3D-printed cuvette; see Methods and Supplementary Fig. 3a for more details. One must carefully eliminate effects of linear dichroism and linear birefringence to assess the true CD signal when the sample under study has structural anisotropy[35,36]. Hence, we adopted a four-configuration measurement approach[14,37,38]; see Methods. Supplementary Figure 3b illustrates this approach, showing measured CD spectra for a highly aligned CNT film under four configurations together with their average. As expected for an aligned film with a racemic mixture of CNTs, there is nearly zero (<0.5 mdeg nm⁻¹) CD observed. A randomly oriented CNT film (Supplementary Fig. 1) and an aligned CNT film prepared using a shear force alignment technique (see Methods) with strong linear optical anisotropy (Supplementary

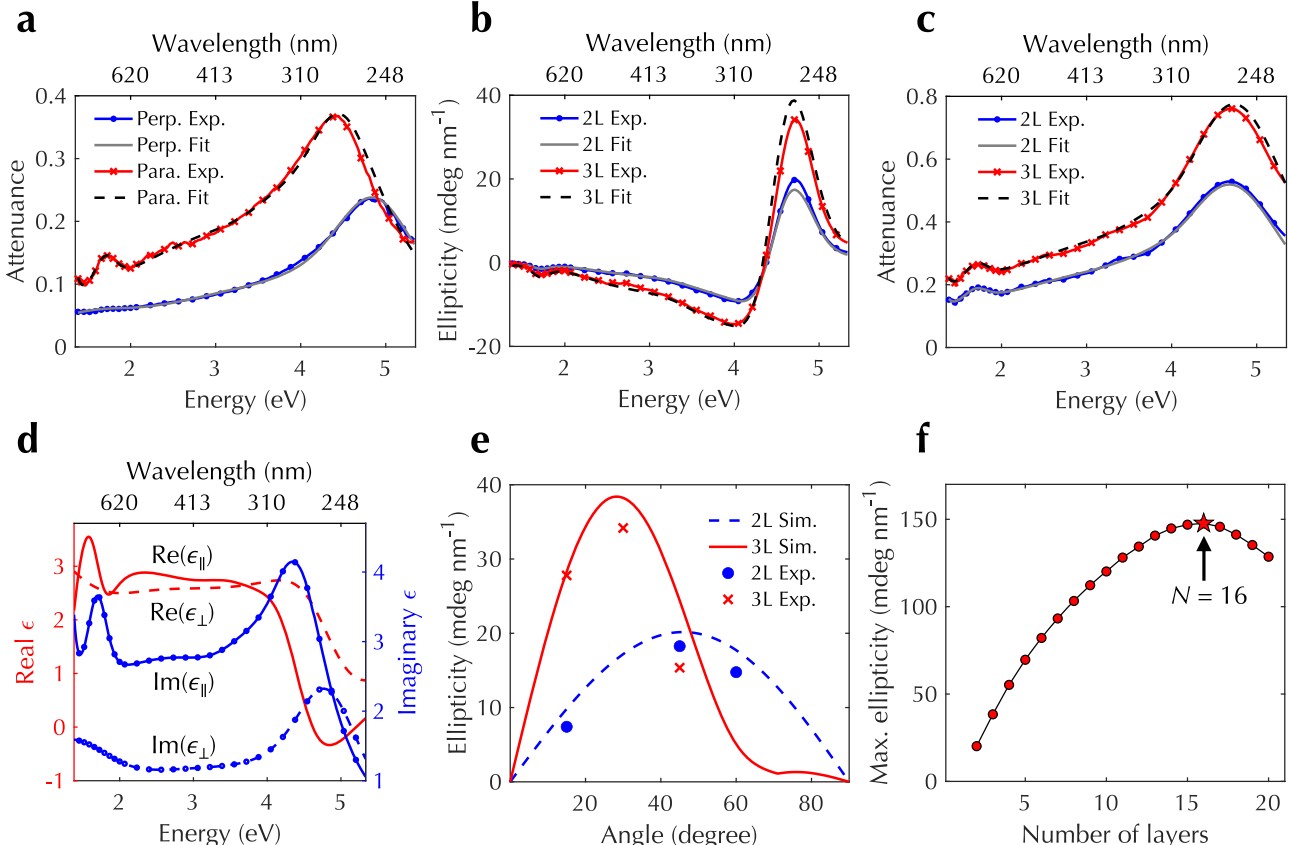

**Fig. 3 | Simulations and analysis of CNT-based synthetic chiral matter using the transfer matrix method.** Experimental **a** linearly polarized attenuation spectra for a highly aligned film (blue and red lines for perpendicular and parallel polarizations, respectively), **b** CD spectra, and **c** unpolarized attenuation spectra for twisted two-layer (blue lines) and three-layer stacks (red lines) with a twist angle of 30°, together with fitting curves (gray solid line and black dashed lines). **d** Extracted complex-valued anisotropic dielectric functions parallel (red lines) and perpendicular (blue lines) to the nanotube axis. The solid lines are real parts and the dashed lines are imaginary parts. **e** Calculated ellipticity as a function of twist angle for twisted two-layer (red line) and three-layer stacks (blue line), which agrees with experimental results (red and blue markers, respectively). **f** Calculated ellipticity as a function of layer number in twisted stacks. Source data are provided as a Source Data file.

Fig. 3c) also show negligible (<0.1 mdeg nm$^{-1}$) CD signals (Supplementary Fig. 3d). To further confirm the validity of this four-configuration measurement approach, we performed spectroscopic ellipsometry measurements and obtained all 16 transmission Müller matrix elements in an energy range of 0.73–6.53 eV. The CD spectrum was obtained based on the differential decomposition of measured Müller matrix[39] (see Methods and Supplementary Fig. 4). Supplementary Figure 5 shows excellent agreement between the spectrum obtained through the four-configuration CD measurements and that from the ellipsometry measurements. In addition to CD, circular birefringence is also noticeable.

The sign of ellipticity can be controlled by the twist direction. Figure 2c displays CD spectra for twisted three-layer stacks with a twist angle of 30° in left-handed and right-handed manners, respectively. We use positive (negative) twist angles for left-handed (right-handed) structures, and the ellipticity has opposite signs at a given photon energy between the left- and right-handed structures. It should be noted that the experimentally recorded ellipticity, 40 ± 1 mdeg nm$^{-1}$, in the three-layer right-handed stack at 4.77 eV is the highest ever obtained in the DUV range. In addition, Supplementary Fig. 6 displays the corresponding $g$-factor spectrum of the three-layer right-handed stack, where the $g$ factor was calculated as the ratio of the differential attenuation (i.e., CD) to the average attenuation of left and right circularly polarized light. The largest $g$ factor is ≈0.07, which is larger than those of natural molecules by 1–2 orders of magnitude and is comparable to those of other existing platforms with large $g$ factors; see Supplementary Table 1 for comparison with other chiral platforms and structures.

Furthermore, we can switch on and off the CD signal by stacking aligned CNT films with twist angles alternating between positive and negative values. Figure 2d shows a series of four spectra, corresponding to one-layer, two-layer, three-layer, and four-layer samples. The one-layer sample shows nearly negligible CD, and the two-layer sample with a twist angle of 30° shows substantial CD, as shown before. We refer to the alignment directions of the first and second layers in the twisted two-layer stack as A and B, respectively. When we stack a third layer with direction A, the obtained architecture corresponds to an ABA configuration in which CD is strongly suppressed, i.e., CD is switched off. When a fourth layer is stacked to construct a configuration of ABAB, CD is restored.

## Transfer matrix calculations

The chiral architectures prepared through the twist-stacking approach described above have well-defined structures that are convenient for analysis using the transfer matrix method. We assumed that the parallel and perpendicular dielectric functions of aligned CNT films can be modeled as a summation of Voigt functions with multiple fitting parameters (see Methods). We then simultaneously fit six experimentally measured spectra: linear-polarization attenuation spectra for a highly aligned CNT film, CD spectra, and unpolarized spectra of twisted two-layer and three-layer stacks with a twist angle of 30°. With such simultaneous fitting, we were able to uniquely determine the fitting parameters and anisotropic dielectric functions. Figure 3a–c shows fitting and experimental spectra, which all demonstrate excellent agreement. Figure 3d displays the

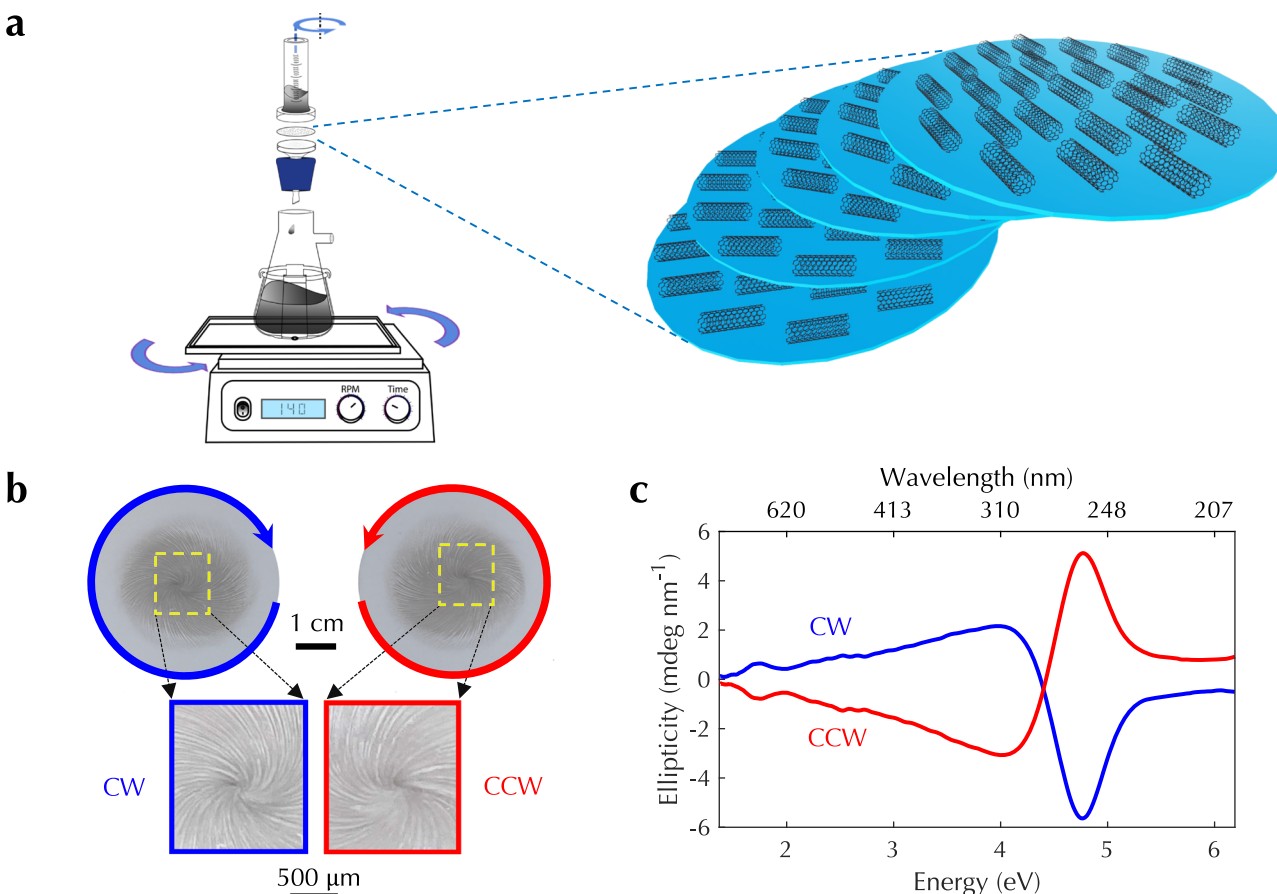

**Fig. 4 | Mechanical-rotation-assisted CVF for creating synthetic chiral matter based on ordered carbon nanotubes. a** Schematic of the mechanical-rotation-assisted vacuum filtration setup. The standard filtration system was mounted on an orbital mechanical shaker, and the rotational motion produced a twisted CNT thin-film architecture. **b** Photograph and **c** corresponding CD spectra of two twisted CNT films produced through mechanical-rotation-assisted CVF under clockwise (CW, blue lines) and counterclockwise (CCW, red lines) rotations. Source data are provided as a Source Data file.

extracted anisotropic complex-valued dielectric functions of the single-layer aligned CNT film.

The obtained dielectric functions and the developed transfer matrix method allow us to understand how the CD signal changes as a function of structure parameters. Figure 3e shows the calculated ellipticity at 4.77 eV for twisted two-layer and three-layer stacks as a function of twist angle. The largest ellipticity occurs at the twist angle $46 \pm 1°$ ($28 \pm 1°$) for a two-layer (three-layer) stack, which agrees well with experimental results in Fig. 2a, b. We also analyzed the twisted two-layer stack using the Jones calculus. The calculation results not only agree with those obtained from the transfer matrix method but also demonstrate that the anisotropic phase response in aligned CNTs is responsible for the observed CD signals in twisted stacks; see Methods and Supplementary Fig. 7 for more details. We further calculated, for an $N$-layer stack ($N = 2 - 20$), the optimal twist angle, i.e., the angle that produces the largest ellipticity, assuming that the twist angle is constant throughout the $N$-layer stack. Figure 3f shows the largest ellipticity as a function of $N$, which exhibits a peak value of 150 mdeg nm$^{-1}$ at $N = 16$. The decreasing ellipticity with increasing $N$ for $N > 16$ can be understood as a result of nearly constant CD and linearly increasing stack thickness with increasing $N$. At the peak, the corresponding $g$ factor is 0.22. Supplementary Figure 8 also displays the optimal twist angle as a function of $N$. In addition to the angles obtained from the transfer matrix method, the optimal angles were also calculated using the analytical half-turn-helix equation $\pi/2N$. Transfer matrix results agree well with those of analytical calculations when $N$ is small, while a deviation between the two becomes noticeable when $N$ increases. This deviation is due to interlayer reflection and

interference effects in multiple layers, which are captured in the transfer matrix method but are not considered in the simple analytical calculations. These analyses offer promising insights into the possibility of creating synthetic CNT-based twisted architectures with stronger chiroptical responses.

## Mechanical-rotation-assisted CVF

The second method we developed is in situ mechanical rotation during CVF, which can convert spontaneous alignment into a wafer-scale twisted structure. As shown in Fig. 4a, the filtration system was placed on an orbital mechanical shaker, which rotated the whole system on a horizontal plane around a vertical axis. Rotational shaking was applied for a short period of time in the middle of the filtration process; see Methods for more details, Supplementary Fig. 9 for an illustration, and Supplementary Movie 1 for an experimental demonstration. During the mechanical-motion-assisted CVF process, individual CNTs are first deposited on the filter membrane to form alignment layers. Mechanical rotation starts afterward and lasts for a certain time duration. While the filtration system is in rotation, a flow field is generated, which controls the alignment orientation of CNTs as CNTs are further deposited to add layers (Fig. 4a). We controlled the rotation speed from 50 to 220 RPM; CNT films fabricated with different morphology are shown in Supplementary Fig. 10. When the rotation speed was high, such as 200 and 220 RPM, strong flow turbulence created inside the funnel led to the formation of nonuniform films, which are less useful for practical applications. When the rotation speed was in an intermediate range of 100–150 RPM, a vortex flow was formed inside the filtration funnel and created a spiral pattern of CNT alignment

orientation, as shown in Fig. 4b. Furthermore, the handedness of the vortex flow was controlled by the direction of mechanical rotation, i.e., clockwise or counterclockwise. The signs of ellipticity around the center of such CNT films produced under opposite mechanical rotation directions were also opposite (Fig. 4c). In addition, CD spectra measured using ellipsometry and four-configuration measurements showed excellent agreement; see Supplementary Figs. 11 and 12.

We further performed detailed spatial mapping of CD spectra in a CNT film by cutting the film into multiple pieces and characterizing piece by piece with a ≈2.2 mm beam size in diameter. Interestingly, we observed different signs of CD at different locations; see Supplementary Fig. 13. Based on our prior work, we know that there are pre-existing aligned long grooves on filter membranes, which determine the alignment direction of CNTs when they are deposited on filter membranes during CVF. Along the lines crossing the film center that are parallel and perpendicular to the direction of the pre-existing grooves, the signs of ellipticity were opposite. This is due to the co-existence of left- and right-handed twisted structures formed between macroscopic alignment layers and spiral rotation layers, as illustrated in Supplementary Fig. 14. Note that, fundamentally, light propagation along the axis of a 3D helical structure is the main reason for the observed chiroptical responses, which was confirmed through finite-difference-time-domain simulations; see Methods and Supplementary Fig. 15. The chiroptical response contribution from a 2D spiral structure is negligibly small compared with that from a 3D helical structure. Furthermore, the fact that both chiralities were observed in a sample with only one rotation handedness and the CD sign spatial distribution within the sample shown in Supplementary Fig. 13 also confirm that the observed chiroptical response mainly originates from the 3D helical structure. Hence, the mechanical–rotation approach can not only create twisted CNT structures faster than the twist-stacking approach but also create both chiralities simultaneously in a single sample, in a single run, and in a controllable manner. In addition to CD mapping, we performed spatial mapping of the reduced linear dichroism (LD$^r$) at 1.88 eV (Methods and Supplementary Fig. 13), as well as mapping of thickness, peak ellipticity, and LD$^r$ at 1.88 eV along the radial direction (Supplementary Fig. 16). Both the thickness and ellipticity were the largest at the center, and thus, center films will be good candidates for applications.

Furthermore, when the mechanical rotation speed was low, such as 50 RPM, the flow vortex was not formed and thus the obtained films were uniform (Supplementary Fig. 10). As a result, the ellipticity sign was constant without any flip within the whole film (Supplementary Fig. 17). This observation further confirms that light propagation along the axis of a 3D helical structure is responsible for chiroptical responses. Finally, although it is challenging to have accurate information on some of the structural parameters (e.g., the thicknesses and angles) of twisted structures formed during mechanical-rotation-assisted CVF, it is theoretically feasible to model obtained films through the effective medium theory by solving wave equations in chiral and bi-isotropic media[40]. The effective parameters describing chiroptical response can be extracted through fitting to experimental data. The in situ mechanical-motion-assisted CVF method creates research opportunities for experimentally controlling and theoretically investigating flow dynamics in the vacuum filtration process to engineer chiroptical response.

## Discussion

Our observation of a high ellipticity of $40 \pm 1$ mdeg nm$^{-1}$ in the DUV range is noteworthy because of the unique properties of this band. For example, most bio- and chemical molecules have large and dispersive refractive indices in the DUV because of their electronic transitions. Thus, high-performance DUV chiral sensors will not only enable ultrasensitive detection of molecules but also differentiate enantiomers[41–45], which can have substantial impacts on pharmaceutical and biomedical science[46,47]. Furthermore, DUV photons in the sunlight are strongly absorbed by the atmosphere so that their propagation is solar-blind and

not affected by the background sun radiation. Hence, incorporating chiral matter systems to construct intrinsically chiral-light-sensitive solar-blind detectors can boost the security and anti-interference capabilities of detectors, which enable functionalities in DUV applications such as missile detection, environment monitoring, non-line-of-sight communication, and astrophysics[48,49].

Furthermore, in the DUV range, the CNT-based chiral platform is advantageous over chiral metamaterials because the available DUV metallic and dielectric materials are limited to only a few candidates, including magnesium[50], aluminum[51–53], and titanium dioxide[54], and the ultrasmall feature sizes and sophisticated geometries of metamaterials make their top-down fabrication challenging. In addition, the demonstrated synthetic chiral matter is fundamentally different from ordinary metamaterials because CNTs are not simple metallic or dielectric rods. Rather, CNTs are quantum wires with quantum confinement, leading to quantized energy levels, or subbands, with energy separation (≈1 eV) much larger than the room-temperature thermal energy (≈26 meV). Optical transitions between such 1D subbands with van Hove singularities in the joint density of states in CNTs manifest themselves as peaks in room-temperature optical spectra. The fact that our observed wavelength-dependent large CD signals are due to these quantum-confinement-induced optical properties enables the engineering of CD signals through the quantum engineering of CNT diameters and atomic structures. The produced architectures represent an alternative type of chiral platform, where millions of densely ordered quantum wires are assembled into a twisted stack. We anticipate that these structures will reveal further non-optical phenomena such as spintronics and attractive possibilities for electronic, photonic, and optoelectronic devices.

## Methods

### Standard CVF
The original CNT powder was made using the arc-discharge approach and was purchased from Carbon Solutions, Inc. with a product number P2 (purity > 90 wt%). In total, 8 mg CNT powder was mixed with 20 mL 0.5% weight concentration sodium deoxycholate aqueous solution and then was dispersed under ultrasonic tip horn sonication for 45 min at 21 W output power. The sonicated suspension was further purified to remove large bundles through ultracentrifugation at 247, $104 \times g$ for 2 h. The supernatant was collected and diluted by 20 times for vacuum filtration. A 2-in. filtration system (MilliporeSigma) and 200-nm-pore-size filter membranes (Whatman Nuclepore Track-Etched polycarbonate hydrophilic membranes, MilliporeSigma) were used. The filtration process was performed in a well-controlled manner, consisting of multiple stages including slow filtration, rapid filtration, and drying processes. Once the polycarbonate filter membrane with CNTs deposited on top was fully dry, the obtained film was transferred onto the desired substrate through a wet transfer process. Specifically, a small droplet of water was first placed on top of the desired substrate. The CNT film was flipped to have the CNT surface to be in contact with the wet substrate and the polycarbonate surface was on top. The water between the CNT film and the substrate was then dried. The top polycarbonate layer was removed by submerging the sample into a chloroform solution. Finally, the sample was cleaned with isopropanol. More details can be found in ref. 20.

### Characterization of highly aligned CNT films
The obtained highly aligned CNT films prepared using standard CVF were characterized using multiple techniques. The linearly polarized optical attenuation spectra were measured using a UV–visible-near-infrared (UV–vis-NIR) spectrometer (Perkin Elmer Lambda 950 UV–vis-NIR) equipped with an automatically controlled rotating polarizer. The measurement energy range was 1.38–6.2 eV, which corresponds to the wavelength range of 200–900 nm. The incident beam diameter was 2.2 mm, which was determined by a customized sample holder as described below. The microscopic structure was characterized using a

scanning electron microscope (FEI TENEO). The surface morphology was characterized using an atomic force microscope (Parksystems NX20). For LD$^r$ mapping, linearly polarized optical attenuation was measured using a 660-nm laser diode with a beam diameter of 2.2 mm, polarizers, and a half-wave plate. Supplementary Fig. 13b displays a polarization-angle-dependent attenuation at 1.88 eV. LD$^r$ was calculated as $2(A_\parallel - A_\perp)/(A_\parallel + A_\perp)$, where $A_\parallel$ and $A_\perp$ are the maximum and minimum attenuation, respectively.

## Twist-stacking of aligned CNT films

The building block for the twist-stacking method was a 14-nm thick 2-in. highly aligned CNT film prepared via standard CVF. The alignment direction of the film was first determined by taking optical images under a microscope to observe the groove direction on the filter membrane[24]. The film was then cut into smaller pieces using a sharp razor blade. After transferring the first film on a fused silica substrate, the whole sample was placed on top of a transparent protractor that was back-illuminated using an LED panel. The second film was then fixed on top of the first one with a rotation angle and transferred using the same wet transfer process. Supplementary Fig. 7 shows the photographs of prepared two-layer and three-layer stacks. Such a process can be repeated for multiple layers with the same or different rotation angles.

## Four-configuration CD measurement

CD spectra were measured using a standard CD spectrometer Jasco J-810 in the energy range of 1.38–6.2 eV, which corresponds to the wavelength range of 200–900 nm. The incident beam diameter was 2.2 mm, which was defined by the sample holder described below. In general, the measured CD signals from solid-state samples (CD$_{measured}$) can be expressed as CD$_{measured}$ = CD$_{true}$ + 0.5(LD′LB − LDLB′) + sin$\alpha$(LD′ sin (2$\theta$) − LDcos(2$\theta$))[38]. LD′ and LB′ are the linear dichroism and linear birefringence for the ±45° reference directions with respect to $xy$ axes[39]. The first term CD$_{true}$ describes the true CD in solid-state samples. The second term 0.5(LD′LB − LDLB′) originates from the interference between macroscopic anisotropies of samples. The third term sin $\alpha$(LD′ sin (2$\theta$) - LDcos(2$\theta$)) comes from the residual static birefringence of the modulator (angle $\alpha$) in the CD spectrometer, which interacts with sample macroscopic anisotropy with the in-plane sample rotation angle $\theta$[38]. The averaging of CD spectra with 90° in-plane rotation can cancel the third term and the averaging of CD spectra with 180° out-of-plane rotation can cancel the second term.

In order to facilitate four-configuration CD measurements with a commercial CD spectrometer, we designed and manufactured a customized sample holder cuvette using stereolithography 3D printing. Supplementary Figure 3 displays the schematic of the designed cuvette main assembly and pocket, which were printed using an Anycubic Mono 3D printer. Two printed parts were then joined together with two bearings and two apertures with a 2.2-mm diameter fixed on both sides of the cuvette to define the light beam size. Finally, we attached copper foils on both sides to block the light passage from other portions. The pocket can rotate 360°. Thus, the solid CNT samples can be measured with the CNT side facing the light beam with varying in-plane rotation angles. The cuvette can be rotated by 180° and the other side of the substrate faces the light beam with varying in-plane rotation angles. As a result, we can obtain CD spectra under four configurations.

## Aligned CNT films fabricated using the shear force technique

To validate the four-configuration CD measurements, we also prepared an aligned CNT film using a shear force technique. Specifically, we used the CNT powder of Meijo DX 302 (purity > 98 wt%), which was also a racemic mixture of semiconducting and metallic CNTs. We used few-walled CNTs (1–2 walls) with an average diameter ≈2 nm, an average aspect ratio ≈3800, and an average length ≈7 μm. The CNT powder was added to chlorosulfonic acid HClSO$_3$ (purity 99 %) and mixed until homogeneously and fully dissolved. The CNTs were aligned by sandwiching the solution between two glass slides, pressing the slides together to remove any air bubbles, applying a shearing force in the direction of the long axis of the slide, and separating the slides. The slides were then placed in a solution of 97 vol% diethyl ether (purity 99%) and 3 vol% oleum (20% fuming with 23.11% free SO$_3$) to remove the chlorosulfonic acid. The slide was removed from the coagulant. The aligned film formed on one glass slide floated on the water by submerging it into water. The floating film was transferred to a fused silica substrate by scooping the film with the substrate and drying the film afterwards[55]. Supplementary Figure 4a confirms the strongly anisotropic absorption in the obtained film and Supplementary Fig. 4b shows the negligible averaged CD signal measured using four-configuration measurements.

## Spectroscopic ellipsometry

The spectra of 16 full transmission Müller matrix elements were measured using an RC2 ellipsometer from J.A. Woollam Company. The measurement was done under normal incidence. The incident beam diameter was ≈2.2 mm adjusted by an aperture in an energy range of 0.73–6.53 eV, which corresponds to a wavelength range of 190–1700 nm. All Müller matrix elements were normalized with respect to the first element $m_{11}$. The obtained Müller matrix is denoted as $M$. The natural logarithm of $M$, which is $L$, consists of CD as shown in the following expression

$$L = \ln(M) = \ln \begin{pmatrix} 1 & m_{12} & m_{13} & m_{14} \\ m_{21} & m_{22} & m_{23} & m_{24} \\ m_{31} & m_{32} & m_{33} & m_{34} \\ m_{41} & m_{42} & m_{43} & m_{44} \end{pmatrix} = \begin{pmatrix} 0 & -LD & -LD' & CD \\ -LD & 0 & CB & -LB' \\ -LD' & -CB & 0 & LB \\ CD & LB' & -LB & 0 \end{pmatrix}.$$

(1)

In addition, we rotated the in-plane polarization every 10° for a full round and we averaged all CD spectra at various in-plane polarization angles so that the influence of imperfect light polarization states in the ellipsometer was negligible. The raw CD signal, obtained as the attenuance difference $\Delta A = A_L - A_R$, was converted into the ellipticity (mdeg) = $\tan^{-1}\{(I_L^{0.5} - I_R^{0.5})/(I_L^{0.5} + I_R^{0.5})\} \approx \Delta A/32980$ (small angle approximation)[25], where $I = I_0 e^{-A\ln10}$ is the transmittance intensity (Beer's law) and $I_0$ is the incident intensity of light. Furthermore, we normalized the ellipticity by the sample thickness to express it in mdeg nm$^{-1}$. At each in-plane polarization angle, the amplitudes of LD, LD′, LB, and LB′ terms were on the order of 10$^{-1}$ while the amplitudes of CD and CB terms were on the order of 10$^{-2}$. During the in-plane sample rotation, LD, LD′, LB, and LB′ terms flipped signs and nearly canceled out after averaging over spectra at all angles, while the CD and CB terms stayed with the same spectral shape (Supplementary Fig. 18).

## Transfer matrix method

A $4 \times 4$ transfer matrix method was used to calculate the optical response of multiple layers of anisotropic non-magnetic materials[56,57]. Here, we detail the $4 \times 4$ transfer-matrix method for calculating the response of anisotropic layered media under normal incidence. This method can avoid the singularity issue for calculating isotropic media under normal incidence using the generalized $4 \times 4$ transfer-matrix method[57]. Specifically, for a material with orthogonal principal axes, such as the directions parallel and perpendicular to CNT alignment, the electromagnetic waves in the $j$th layer are expressed as

$$\begin{pmatrix} \mathbf{E}_j \\ \mathbf{H}_j \end{pmatrix} = \sum_{\sigma=1}^{4} E_{j,\sigma} \begin{pmatrix} \mathbf{e}_{j,\sigma} \\ \mathbf{h}_{j,\sigma} \end{pmatrix} \exp(ik_{j,\sigma}z - i\omega t),$$

(2)

$$\mathbf{h}_{j,\sigma} = \frac{k_{j,\sigma}\hat{\mathbf{z}} \times \mathbf{e}_{j,\sigma}}{\omega\mu_0},$$

(3)

$$k_{j,1} = -k_{j,2} = k_0 n_{s_j}, \quad (4)$$

$$k_{j,3} = -k_{j,4} = k_0 n_{p_j}, \quad (5)$$

$$\mathbf{e}_{j,1} = \mathbf{e}_{j,2} = \sin\theta_j \hat{\mathbf{y}} + \cos\theta_j \hat{\mathbf{x}}, \quad (6)$$

$$\mathbf{e}_{j,3} = \mathbf{e}_{j,4} = \cos\theta_j \hat{\mathbf{y}} - \sin\theta_j \hat{\mathbf{x}}, \quad (7)$$

$$\mathbf{h}_{j,1} = -\mathbf{h}_{j,2} = \frac{k_0 n_{s_j}}{\omega\mu_0}(-\sin\theta_j\hat{\mathbf{x}} + \cos\theta_j\hat{\mathbf{y}}), \quad (8)$$

$$\mathbf{h}_{j,3} = -\mathbf{h}_{j,4} = \frac{k_0 n_{p_j}}{\omega\mu_0}(-\cos\theta_j\hat{\mathbf{x}} - \sin\theta_j\hat{\mathbf{y}}), \quad (9)$$

where $n_{s_j}$ is the refractive index along the $s_j$ axis of the $j$-th layer, $n_{p_j}$ is the refractive index along the $p_j$ axis of the $j$-th layer, $\theta_j$ is the angle of the $s_j$ axis with respect to the $x$-axis in counterclockwise rotation, $k_0$ is vacuum wavevector, $z$ is the propagation distance along $z$-axis, $\omega$ is angular frequency, and $\hat{\mathbf{x}}$, $\hat{\mathbf{y}}$, and $\hat{\mathbf{z}}$ are unit vectors along $x$-, $y$-, and $z$-axes, respectively.

The dielectric function tensor $\epsilon_j$ of the $j$-th layer in $xy$ coordinate systems can be written as

$$\epsilon_j = R_j \begin{pmatrix} \epsilon_{s_j} & 0 \\ 0 & \epsilon_{p_j} \end{pmatrix} R_j^{-1}, \quad (10)$$

$$R_j = \begin{pmatrix} \cos\theta_j & -\sin\theta_j \\ \sin\theta_j & \cos\theta_j \end{pmatrix}. \quad (11)$$

Applying the boundary conditions that tangential components of **E** and **H** are continuous at the boundary of each layer, we obtain

$$\begin{pmatrix} E_{j+1,1} \\ E_{j+1,2} \\ E_{j+1,3} \\ E_{j+1,4} \end{pmatrix} = D_{j+1}^{-1} D_j P_j \begin{pmatrix} E_{j,1} \\ E_{j,2} \\ E_{j,3} \\ E_{j,4} \end{pmatrix} = M_{j+1,j} P_j \begin{pmatrix} E_{j,1} \\ E_{j,2} \\ E_{j,3} \\ E_{j,4} \end{pmatrix} \quad (12)$$

with

$$D_j = \begin{pmatrix} \sin\theta_j & \sin\theta_j & \cos\theta_j & \cos\theta_j \\ -n_{s_j}\sin\theta_j & n_{s_j}\sin\theta_j & -n_{p_j}\cos\theta_j & n_{p_j}\cos\theta_j \\ n_{s_j}\cos\theta_j & -n_{s_j}\cos\theta_j & -n_{p_j}\sin\theta_j & n_{p_j}\sin\theta_j \\ \cos\theta_j & \cos\theta_j & -\sin\theta_j & -\sin\theta_j \end{pmatrix} \quad (13)$$

and

$$P_j = \begin{pmatrix} e^{ik_{j,1}d_j} & 0 & 0 & 0 \\ 0 & e^{ik_{j,2}d_j} & 0 & 0 \\ 0 & 0 & e^{ik_{j,3}d_j} & 0 \\ 0 & 0 & 0 & e^{ik_{j,4}d_j} \end{pmatrix}, \quad (14)$$

where $d_j$ is the thickness of the $j$-th layer.

The amplitude of electromagnetic components in the last ($N$th) layer can be related to the amplitude in the first layer by multiplying all

matrices as

$$\begin{pmatrix} E_{1,N} \\ E_{2,N} \\ E_{3,N} \\ E_{4,N} \end{pmatrix} = M_{N,N-1} P_{N-1} M_{N-1,N-2} P_{N-2} \cdots M_{2,1} P_1 \begin{pmatrix} E_{1,1} \\ E_{2,1} \\ E_{3,1} \\ E_{4,1} \end{pmatrix} = Q \begin{pmatrix} E_{1,1} \\ E_{2,1} \\ E_{3,1} \\ E_{4,1} \end{pmatrix}. \quad (15)$$

$Q$ is the transfer matrix of the whole system, from which we can calculate mode-dependent complex transmission and reflection coefficients. Mode indices 1–4 correspond to $s$-wave forward, $s$-wave backward, $p$-wave forward, and $p$-wave backward modes, respectively. The transmitted field ($E_{t,s_N}, E_{t,p_N}$), reflected field ($E_{r,s_N}, E_{r,p_N}$), and the incident field ($E_{i,s_N}, E_{i,p_N}$) can be related through

$$\begin{pmatrix} E_{t,s_N} \\ 0 \\ E_{t,p_N} \\ 0 \end{pmatrix} = Q \begin{pmatrix} E_{i,s_1} \\ E_{r,s_1} \\ E_{i,p_1} \\ E_{r,p_1} \end{pmatrix} = \begin{pmatrix} Q_{11} & Q_{12} & Q_{13} & Q_{14} \\ Q_{21} & Q_{22} & Q_{23} & Q_{24} \\ Q_{31} & Q_{32} & Q_{33} & Q_{34} \\ Q_{41} & Q_{42} & Q_{43} & Q_{44} \end{pmatrix} \begin{pmatrix} E_{i,s_1} \\ E_{r,s_1} \\ E_{i,p_1} \\ E_{r,p_1} \end{pmatrix}. \quad (16)$$

As a result, we can derive reflection and transmission coefficients for $s$- and $p$-waves in terms of the matrix elements of $Q$ as

$$r_{ss} = \left.\frac{E_{r,s_1}}{E_{i,s_1}}\right|_{E_{i,p_1}=0} = \frac{Q_{24}Q_{41} - Q_{21}Q_{44}}{Q_{22}Q_{44} - Q_{24}Q_{42}}, \quad (17)$$

$$r_{sp} = \left.\frac{E_{r,p_1}}{E_{i,s_1}}\right|_{E_{i,p_1}=0} = \frac{Q_{21}Q_{42} - Q_{22}Q_{41}}{Q_{22}Q_{44} - Q_{24}Q_{42}}, \quad (18)$$

$$t_{ss} = \left.\frac{E_{t,s_N}}{E_{i,s_1}}\right|_{E_{i,p_1}=0} = Q_{11} + \frac{Q_{12}(Q_{24}Q_{41} - Q_{21}Q_{44}) + Q_{14}(Q_{21}Q_{42} - Q_{22}Q_{41})}{Q_{22}Q_{44} - Q_{24}Q_{42}} \quad (19)$$

$$t_{sp} = \left.\frac{E_{t,p_N}}{E_{i,s_1}}\right|_{E_{i,p_1}=0} = Q_{31} + \frac{Q_{32}(Q_{24}Q_{41} - Q_{21}Q_{44}) + Q_{34}(Q_{21}Q_{42} - Q_{22}Q_{41})}{Q_{22}Q_{44} - Q_{24}Q_{42}}. \quad (20)$$

$$r_{ps} = \left.\frac{E_{r,s_1}}{E_{i,p_1}}\right|_{E_{i,s_1}=0} = \frac{Q_{24}Q_{43} - Q_{23}Q_{44}}{Q_{22}Q_{44} - Q_{24}Q_{42}}, \quad (21)$$

$$r_{pp} = \left.\frac{E_{r,p_1}}{E_{i,p_1}}\right|_{E_{i,s_1}=0} = \frac{Q_{23}Q_{42} - Q_{22}Q_{43}}{Q_{22}Q_{44} - Q_{24}Q_{42}}, \quad (22)$$

$$t_{pp} = \left.\frac{E_{t,s_N}}{E_{i,p_1}}\right|_{E_{i,s_1}=0} = Q_{33} + \frac{Q_{32}(Q_{24}Q_{43} - Q_{23}Q_{44}) + Q_{34}(Q_{23}Q_{42} - Q_{22}Q_{43})}{Q_{22}Q_{44} - Q_{24}Q_{42}}, \quad (23)$$

$$t_{ps} = \left.\frac{E_{t,p_N}}{E_{i,p_1}}\right|_{E_{i,s_1}=0} = Q_{13} + \frac{Q_{12}(Q_{24}Q_{43} - Q_{23}Q_{44}) + Q_{14}(Q_{23}Q_{42} - Q_{22}Q_{43})}{Q_{22}Q_{44} - Q_{24}Q_{42}}. \quad (24)$$

For layers with isotropic media, such as the input and output air layers, the principal axes are chosen to be $s_1 = s_N = \hat{\mathbf{x}}$ and $p_1 = p_N = \hat{\mathbf{y}}$.

In addition, the dielectric functions parallel and perpendicular to the CNT alignment direction from the near-infrared to UV ranges were

modeled as a summation of Voigt functions[58]

$$\epsilon_{s,p}(\omega) = \epsilon_{\infty,s,p} + \sum_{n=1}^{N} C_{V,s,p}(\omega)\Big|_{A_n,\omega_{0,n},\gamma_{L,n},\gamma_{G,n}} \qquad (25)$$

with

$$C_V(\omega)\Big|_{A,\omega_0,\gamma_L,\gamma_G} = -A\frac{\mathrm{Im}(F(x-x_0-iy)+F(x+x_0+iy))}{\mathrm{Re}(F(iy))} \qquad (26)$$

$$+iA\frac{\mathrm{Re}(F(x-x_0+iy)-F(x+x_0+iy))}{\mathrm{Re}(F(iy))} \qquad (27)$$

and

$$x = \frac{2\sqrt{\ln 2}}{\gamma_G}\omega, x_0 = \frac{2\sqrt{\ln 2}}{\gamma_G}\omega_0, y = \frac{\gamma_L\sqrt{\ln 2}}{\gamma_G}, \qquad (28)$$

where $\omega$ is angular frequency, $A$ is amplitude factor, $\omega_0$ is resonance frequency, $\gamma_L$ is Lorentz linewidth, $\gamma_G$ is Gaussian linewidth, and $F$ is Faddeeva function. $N$ is selected as 5. For each polarization ($s$ or $p$), there are 21 fitting parameters, including $\{A_n, \omega_{0,n}, \gamma_{L,n}, \gamma_{G,n}, n = 1-5\}$ and $\epsilon_\infty$. There are in total 42 fitting parameters for both polarizations.

## Jones calculus for the twisted two-layer stack

Supplementary Figure 8a displays the schematic of a twisted two-layer stack of aligned CNTs with defined coordinates. The complex-valued field transmission coefficients parallel and perpendicular to the tube axis are denoted as $t_\parallel$ and $t_\perp$, respectively. The twist angle between two layers is denoted as $\theta$. Compared to the transfer matrix method, the analysis of twisted stacks using the Jones calculus is a simplified approach without considering multiple reflections between layers. For two-layer stacks, the Jones calculus analysis is a good approximation. Specifically, the Jones matrix for the first layer of aligned CNTs can be expressed as

$$J_{\mathrm{CNT}} = \begin{pmatrix} t_\parallel & 0 \\ 0 & t_\perp \end{pmatrix} \qquad (29)$$

in the $x-y$ coordinate. The rotation matrix to convert the $xy$ coordinate to the $x'y'$ coordinate is

$$J_{\mathrm{rot}} = \begin{pmatrix} \cos\theta & -\sin\theta \\ \sin\theta & \cos\theta \end{pmatrix}. \qquad (30)$$

Hence, the Jones matrix of the second layer of aligned CNTs is the same as that of the first layer in the $x'y'$ coordinate. The input circularly polarized light of different handedness in the $xy$ coordinate can be expressed as vectors

$$\mathbf{E}_{\mathrm{in}} = \begin{pmatrix} 1 \\ \pm i \end{pmatrix}, \qquad (31)$$

respectively.

As a result, the output vector in the $x'y'$ coordinate is

$$\mathbf{E}_{\mathrm{out}} = \begin{pmatrix} E_{x'} \\ E_{y'} \end{pmatrix} = J_{\mathrm{CNT}} J_{\mathrm{rot}} J_{\mathrm{CNT}} \mathbf{E}_{\mathrm{in}}, \qquad (32)$$

and the output intensity is

$$I = (E_{x'}^*, E_{y'}^*)\begin{pmatrix} E_{x'} \\ E_{y'} \end{pmatrix}, \qquad (33)$$

where $E_{x'}^*$ ($E_{y'}^*$) is the complex conjugate of $E_{x'}$ ($E_{y'}$). For the $(1, i)^T$ input, the output intensity $I_1$ is calculated as

$$\begin{aligned}|t_\perp|^4\cos^2\theta + 2|t_\parallel|^2|t_\perp|^2\sin^2\theta + |t_\parallel|^4\cos^2\theta \\ + |t_\perp||t_\parallel|(|t_\perp|^2 - |t_\parallel|^2)\sin(2\theta)\sin\phi,\end{aligned} \qquad (34)$$

where $\phi$ is the phase difference between $t_\parallel$ and $t_\perp$. Similarly, for the $(1, -i)^T$ input, the output intensity $I_2$ is calculated as

$$\begin{aligned}|t_\perp|^4\cos^2\theta + 2|t_\parallel|^2|t_\perp|^2\sin^2\theta + |t_\parallel|^4\cos^2\theta \\ - |t_\perp||t_\parallel|(|t_\perp|^2 - |t_\parallel|^2)\sin(2\theta)\sin\phi.\end{aligned} \qquad (35)$$

Hence, the difference between $I_1$ and $I_2$ as a function of twist angle follows a $\sin(2\theta)$ dependence. As shown in Supplementary Fig. 7b, the calculated ellipticity as a function of twist angle using the transfer matrix method can be well fit with a $\sin(2\theta)$ function. The largest difference between $I_1$ and $I_2$ occurs when $\sin(2\theta) = 1$ and thus $\theta = 45°$, which also agrees with our experimental results. In addition, non-zero $\phi$ is required to produce the difference between $I_1$ and $I_2$, indicating that the anisotropic phase response in aligned CNTs is responsible for observed CD signals.

## Mechanical-rotation-assisted CVF

The filtration system was mounted on top of an orbital shaker (Lab Companion OS-2000) using a 100 mL flask clamp. The overall system was rotated in a circle with a diameter of 19.1 mm. The rotation speed was controllable in a range from 20 to 500 RPM, and the rotation duration was 30 s.

## FDTD simulations

The FDTD simulations were done with Anasys Lumerical software. The dielectric function tensor $\epsilon$ of aligned CNT films was from Fig. 3d. When the alignment orientation was rotated, the dielectric function tensor became

$$R\epsilon R^{-1}, \text{ with } R = \begin{pmatrix} \cos\theta & -\sin\theta \\ \sin\theta & \cos\theta \end{pmatrix}, \qquad (36)$$

where $\theta$ is the rotation angle. Bloch boundary conditions were used in the sample plane.

## Reporting summary

Further information on research design is available in the Nature Portfolio Reporting Summary linked to this article.

## Data availability

The data that support the findings of this study are available from the corresponding author upon request. Source data are provided in this paper.

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

## Acknowledgements

We thank Bruce Weisman, Tonya Cherkuri, and Somesh Sasmal for useful discussions. We also thank the staff of the University of Utah Nanofab, Electron Microscopy and Surface Analysis Lab, and Rice Shared Equipment Authority for technical assistance. J.D., M.L. and W.G. acknowledge support from the University of Utah startup fund. W.G. acknowledges support from the National Science Foundation through Grant No. 2321366. J.D. and J.K. acknowledge support from the Robert A. Welch Foundation through Grant No. C-1509, the Air Force Office of Scientific Research through Grant No. FA9550-22-1-0382, and the Chan Zuckerberg Initiative through Grant No. WU-21-357. J.K. and W.G. acknowledge support from the National Science Foundation through Grant No. 2235276. O.D. and M.P. acknowledge support from the Department of Energy through Grant No. DE-AR0001015 (Advanced Research Projects Agency-Energy) and Robert A. Welch Foundation through Grant No. C-1668. K.Y. acknowledges support from JSPS KAKENHI through Grants No. JP20H02573, JP21H05017, JP22H05469, JP23H00259, and JST CREST through Grant No. JPMJCR17I5. J.F., M.P., J.K., W.G. (U.S. side), and Y.Y., K.Y., R.S. (Japan side) acknowledge support from the PIRE U.S.-Japan Program through Grants No. 2230727 (National Science Foundation, U.S.) and JPJSJRP20221202 (JSPS, Japan). R.S. acknowledges support from JSPS KAKENHI through Grant No. JP22H00283 and Yushan Fellow Program from Taiwan.

## Author contributions

W.G. conceived the idea, designed experiments, and supervised the project. J.D. performed the experiments with the help of A.B. and K.Z. and under the support and guidance of J.K. and W.G. M.L. conducted theoretical modeling and calculations with the help of J.F. and under the guidance of W.G. O.D. and Y.Y. helped with the preparation of samples under the support of M.P. and K.Y., respectively. N.H. performed Müller matrix spectroscopic ellipsometry measurements. R.S. contributed to the explanation of experimental observations. All authors discussed the results and contributed to the paper.

## Competing interests

The authors declare no competing interests.
