## [Peer Review File · Nature Communications]

Engineering Chirality at Wafer Scale with Ordered Carbon Nanotube ArchitecturesREVIEWER COMMENTS

Reviewer #1 (Remarks to the Author):

This manuscript by Doumani et al. reports two approaches (namely, in situ mechanical rotation and manual twisted stacking) for realizing artificial chirality in carbon nanotube (CNT) assemblies. The fabricated CNT films are wafer-sized, ultrathin, and possess excellent chiroptical properties characterized by their circular dichroism (CD), which is important for the field of chiral photonics and optoelectronics. For twist-stacked CNTs, the CD spectral features are qualitatively recapitulated using transfer matrix method calculations. A thickness-normalized ellipticity of up to ~ 40 deg/ μm in the deep ultraviolet is experimentally observed (and an even higher one of ~ 150 deg/ μm for calculated multilayer stacks), which is impressive for UV chiral nanostructures and metamaterials. The work is interesting and the chiroptical performance of the nanoassemblies is good. However, the current study lacks a conceptual advance and detailed analyses, especially for the first method on mechanical-rotation-assisted CVF. Therefore, I would like to reconsider the manuscript for publication in Nature Communications contingent upon major revisions from the authors. Please see questions/suggestions below:

1. In the Abstract, the authors claimed that the macroscopic optical properties of the twist-stacked CNT films are governed by van Hove singularities (vHS). There is no data supporting any quantum mechanical effects considering there is no e.g., diameter or chirality sorting of the CNTs. I am also failing to see how the CD signal is a result of 1D vHS (page 9, line 400).
2. Fig1c: could the authors explain the polarization-dependency of the absorption spectra of aligned CNT? Why did the S22 and M11 transitions disappeared in the perpendicular case? And why is there an energy shift in the π -plasmon resonance?
3. For the in situ mechanical rotation method, CNTs assemblies appear to have a spiral configuration (Fig 2a inset) that is responsible for its chiroptical activity. The author claimed that the CNTs assemble into a spiral castle (page 9, line 403). I personally find it interesting, but a systematic investigation is missing. I would like the authors to provide further information (e.g., structural characterization) on the assembly formation during the rotational motion and the processing factors affecting this chiral assembly.
4. The authors should provide an AFM height profile to determine the thickness of the spiral CNT assembly per Fig 2. I would assume that this is different from the aligned anisotropic assembly per Fig 1.
5. For the in situ mechanical rotation method, quantitative processing-structure-chiral property relationships should be provided. For instance, what is different for the assembly after 70s-rotation that leads to a large ellipticity value? What is the effect of the strength of the rotational force on the chiroptical response of the assembly?
6. Artificial twisted stacking as a strategy has been applied to various nanostructures to elicit artificial chirality, beyond the chiral bilayer graphene case the authors mentioned (Kim et al. Nat. Nanotechnol. 2016, 11, 520), i.e., the method itself is not conceptually novel. Some more exemplary systems and recent works/reviews are missing such as (1) Han et al. Adv. Mater. 2023, 35, 2206141; (2) Lv et al. Angew. Chem. Int. Ed. 2017, 56, 5055 –5060; (3) Wu et al. Nano Lett. 2021, 21, 8298–8303; etc. The authors should better describe these and balance their references accordingly.

Reviewer #2 (Remarks to the Author):

The publication “controlled synthetic chirality in macroscopic assemblies of carbon nanotubes” by Doumani et al. reports the fabrication of large-area chiral thin films made of twisted oriented layers of carbon nanotubes. The films display tunable chiroptical properties in the UV and visible range. While it has already been demonstrated that twisted stacking of oriented 1D nanoparticles lead to strong tunable circular dichroism, it is the first time that such a structure is produced with carbon nanotubes

to study their optical properties. A big advantage of the structure is that it shows chiroptical properties in the deep UV. Overall, this paper is well written, and the figures are clear and well structured. Therefore, I recommend publication after addressing my comments listed below:

1. Although the alignment technique has already been published before (ref 13 and 14 in the manuscript), I suggest that the authors give a short summary of the mechanism that leads to the orientation of the CNT over such large areas.
2. There are many other examples of similar structures of twisted layers of oriented 1D nano-objects. Twisted layers of CNT have for instance already been prepared for their enhanced mechanical properties (DOI: 10.1002/adma.201605750). The chiroptical properties of Bouligand nanostructures was also widely studied. This structure is for instance found to occur naturally in the cuticle of some beetles (doi.org/10.1364/AO.49.004558) or in films of cellulose nanocrystals (DOI: 10.1002/anie.201500277), and has been synthetically prepared with metallic oxide nanowires (DOI: 10.1002/anie.201903264) or metallic nanowires/nanorods (see for instance doi.org/10.1021/acsnano.1c04804, 10.1021/acs.nanolett.1c02812 or DOI: 10.1038/ncomms1877). This should be better highlighted in the introduction, and the optical properties of the CNT films should be compared to the other systems.
3. The sample obtained with mechanical-rotation-assisted CVF shows a spiral-like pattern on the substrate. Is the CD measurement done in the center of this spiral or away from the center? Is the CD spectrum dependent on the position at which the measurement is done? I think that the RC2 Mueller ellipsometer that was used can do mapping of the sample, I suggest that the authors add such a large-area mapping of the CD over the substrate to probe the homogeneity of the chiroptical properties. As discussed in the text, linear dichroic effects are quite large in such samples. It would be good to show a mapping of the LD over the sample along with CD mapping. Furthermore, is the local orientation of the CNT homogeneous along the thickness of the film (parallel stacking of CNT when looking perpendicular to the film)?
4. Fig. 2c and Fig. 2d show reversed CD spectra. Were those measurements done on samples with a different handedness, or the sample was simply reversed (as shown in Fig. 2b). This information should be added in the text or in the caption. Furthermore, is the CD spectrum reversed when the orbital shaker is rotated in the opposite direction?
5. A lot of studies rely on the g-factor ($g = \Delta\epsilon/\epsilon$) to compare the intrinsic chiroptical properties of different samples. Authors have made the choice to use the ellipticity normalized to the thickness, which also makes sense. It would however be worthwhile to also show (at least) one g-factor spectrum and compare to the values typically obtained in the literature. Indeed, depending on the targeted application, it is important to know if the structure is lossy (absorption) or if the CD arises in a transparent film based only on differences in the refractive index for LCP and RCP light (phase-retardation effect).
6. The modeling by the Transfer Matrix approach is only shown for samples prepared by the twist-stacking of multiple aligned CNT films. Is it also possible to model the samples obtained by mechanical-rotation-assisted CVF?
7. It is shown that the CD is maximal for 3 layers oriented at 30°. What happens if 3 layers are oriented with a twist angle of 60°? This would be equivalent to interchange the second and third layer in a 3x30° sample, and I wonder if the CD spectrum is modified by doing so. Furthermore, what happens when more than 3 layers are stacked with a twist angle of 30°. In other words, is the maximal CD obtained for a half turn of the "helix" or does the CD increases further when more layers are added?
8. At the end of the conclusion (p. 9 l. 402), authors mention that "these structures can potentially reveal new phenomena, including non-optical phenomena...". It should be specified which non-optical phenomena could be revealed.

Reviewer #3 (Remarks to the Author):

This is a very interesting paper, describing a method for preparing thin films of commercial carbon nanotubes (CNT) with outstanding chiroptical signatures. The experimental results are convincing and robust. They display the potentiality of the methods. They are also matched well by transfer matrix calculations. This success must arise also from the particularly straightforward spectroscopy of the CNT. With these tools at hand, the authors designed some crucial experiments to demonstrate the major features of their approach (rotatory filtration vs. skewed film transfer; twist angle, layer thickness. The experimental protocols are well described and appear to enable other researchers to replicate the experiments described herein. The manuscript is very readable and it refers to literature in a balanced and complete way.

For all these reasons, I recommend publication.

Two tiny suggestions. I notice that there is a mixture of energy and wavelengths in the text. In some cases both are reported, in some other only one. Please make your own choice. It would be very useful to have both indicated in the horizontal scale (for example eV at bottom and λ at the top) of the plots.

Especially in the legend of figure 2, it is not really clear that the ellipticity is normalized to the film thickness.

Reviewer #4 (Remarks to the Author):

Remarks to the Author

In this manuscript the authors report on structural circular dichroism (CD) in assemblies of carbon nanotubes (CNT) fabricated by two methods based on controlled vacuum filtration (CVF): i) mechanical rotation during CVF and ii) twist-stacking of aligned CNT films produced by CVF. This study is interesting for the development of simple methods to fabricate chiral systems. There are some questions/concerns for the authors to consider.

1. In method i), very good agreement of CD spectra determined by standard CD spectroscopy and differential decomposition of the Mueller matrix is reported (figure 2c). However, the authors should discuss the effects of sample inhomogeneity, which is clearly noticed in figure 2a. Sample homogeneity is critical for all possible applications referred in the text.
2. Do data in figure 2c are from the center of the sample?
3. How LD, LD', LB, LB', CB, and CD look along a radius?
4. In supplementary figure 5, averaged polarization properties from Mueller matrices measured at rotation angles between 0° and 360° are reported. It is noticed that values of $\pm 45^\circ$ linear diattenuation (LD') and $\pm 45^\circ$ linear birefringence (LB') are of the same order than CD and circular birefringence (CB). It should be discussed the physical meaning of these results.
5. Regarding the twist-stacking method ii), appreciable values of linear diattenuation and linear birefringence are expected. The differential decomposition of the Mueller matrix of two-layer and three-layer stacks with a twist angle of 30° should be reported to demonstrate the equivalence with CD spectra shown in figure 4b.

We thank all referees for their careful review of our manuscript and thoughtful comments. Below we address each of the questions/comments in detail:

Response to Reviewer #1

Reviewer #1's comment #1: In the Abstract, the authors claimed that the macroscopic optical properties of the twist-stacked CNT films are governed by van Hove singularities (vHS). There is no data supporting any quantum mechanical effects considering there is no e.g., diameter or chirality sorting of the CNTs. I am also failing to see how the CD signal is a result of 1D vHS (page 9, line 400).

Response to comment #1: Based on our experimental results, which quantitatively agree with our simulation results, we can conclude that the observed deep ultraviolet (DUV) circular dichroism (CD) signals in twist-stacked carbon nanotube (CNT) films arise from the interaction of DUV light with the rotated dipole oscillators of CNT DUV absorption features (“ π ” peaks in **Fig. 1c** of the revised manuscript) along the light propagation direction.

The strong circumferential quantum confinement on the order of ~ 1 nm in CNTs leads to the formation of subbands and van Hove singularities (vHS) in CNT density of states that are characteristic of a one-dimensional (1D) system. The quantum confinement is so strong that the energy separation of subbands (~ 1 eV) is much larger than the room-temperature thermal energy (26 meV at 300 K). Hence, the interband optical features, including DUV absorption features, in room-temperature CNT spectra are from the (excitonically enhanced) resonant transitions between 1D subbands that occur at vHS.

Specifically, DUV spectral features are due to the interband transitions in the vicinity of the M point of the Brillouin zone of graphene, which is also dominated by CNT quantum confinement. The diameter-dependent and chirality-dependent DUV spectral features have been theoretically and experimentally demonstrated. For example, Takagi and Okada have calculated CNTs' DUV spectral features based on density functional theory and the tight-binding approximation, revealing strong diameter and chirality dependence (*Takagi and Okada, "Theoretical calculation for the ultraviolet optical properties of single-walled carbon nanotubes," Phys. Rev. B 79, 233406, (2019)*, which is the Ref. [27] of the revised manuscript). Furthermore, such diameter and chirality dependence has also been experimentally observed; see, e.g., *Ranch et al., "UV-vis absorption spectroscopy of carbon nanotubes: Relationship between the π -electron plasmon and nanotube diameter," Chemical Physics Letters 493, 19-23, (2010)*, and *Liu et al., "Large-scale single-chirality separation of single-wall carbon nanotubes by simple gel chromatography," Nat. Comm. 2, 309 (2011)*. Note that CNT “chirality” here refers to CNTs' atomic structures instead of the handedness (i.e., enantiomer) of CNTs.

In summary, the observed DUV CD signals in twist-stacked CNTs originate from characteristic features of the 1D band structure of CNTs. Such features are determined by the diameters and chiralities of the CNTs involved and are well understood from previous extensive optical studies of CNTs. This fact profoundly differentiates our work on twist-stacked CNT films from prior work on twist-stacked layers of conventional materials (e.g., silver and gold nanorods). The optical properties of the latter are governed by bulk material properties; see also our response to the reviewer's comment #6. Furthermore, our work opens new opportunities for creating and engineering chiral matter based on 1D quantum wires (not limited to CNTs) for exploring novel chiral photonic and optoelectronic devices and beyond.

In the revised manuscript, we have added new sentences and references to further explain CNT's fundamental optical properties. The detailed changes are as follows:

Changes Made:

Main text, Line 191 - 194

Added: Specifically, the DUV spectral features originate from interband transitions associated with the M point of the graphene Brillouin zone [27, 28], and the transition energies depend on the diameters and atomic structures of CNTs [27, 29, 30].

Main text, Line 482 – 490

Added red texts: Rather, CNTs are quantum wires with extreme quantum confinement, leading to quantized energy levels, or subbands, with energy separation (~ 1 eV) much larger than the room-temperature thermal energy (26 meV). Optical transitions between such 1D subbands with van Hove singularities in the joint density of states in CNTs manifest themselves as peaks in room-temperature optical spectra. The fact that our observed wavelength-dependent giant CD signals are due to these quantum-confinement-induced optical properties enables the engineering of CD signals through the quantum engineering of CNT diameters and atomic structures.

Reviewer #1's comment #2: Fig1c: could the authors explain the polarization-dependency of the absorption spectra of aligned CNT? Why did the S22 and M11 transitions disappeared in the perpendicular case? And why is there an energy shift in the π -plasmon resonance?

Response to comment #2: The strongly polarization-dependent optical properties of aligned CNTs arise from the optical selection rules of CNTs associated with 1D electronic states with well-defined angular momenta. Light polarized parallel to the CNT axis has no angular momentum component along the CNT axis, so only the angular-momentum-preserving transitions, that is, only transitions between subbands with the same subband indices are allowed, such as S₂₂ and M₁₁; see **Fig. R1C2**. On the other hand, light polarized perpendicular to the CNT axis is a 50%-50% mixture of circularly polarized light with opposite helicity carrying angular momenta of ± 1 , which can induce transitions between subbands with an index difference of ± 1 , such as S₁₂ and M₁₂; see **Fig. R1C2**. However, because of the so-called depolarization effect, perpendicular transitions are strongly suppressed compared to parallel transitions. Our prior studies, Yanagi *et al.*, "Intersubband plasmons in the quantum limit in gated and aligned carbon nanotubes," *Nat. Comm.* 9, 1121 (2018) and Katsutani *et al.*, "Direct observation of cross-polarized excitons in aligned single-chirality single-wall carbon nanotubes," *Phys. Rev. B* 99, 035426 (2019), have more detailed descriptions of CNT optical selection rules in aligned CNTs.

Regarding DUV spectral features, the difference in resonance peaks (i.e., energy shift) between parallel and perpendicular polarization is believed to be due to the excitation of different interband transitions in CNTs. However, the nature of such transitions (e.g., excitonic or plasmonic) has not been clarified, and no systematic studies exist that can provide clear assignments to these transitions. Detailed studies of polarization-dependent spectroscopy of aligned CNTs with various diameters and chiralities are needed, but such studies have been precluded due to the lack of appropriate samples. In fact, highly aligned CNT films fabricated through our vacuum filtration technique offer unique future opportunities to provide a clear-cut understanding of the origins of these DUV features in CNTs.

Fig. R1C2: Band structures of metallic and semiconducting CNTs.

In the revised manuscript, we have added new sentences and references to further explain CNT's fundamental optical properties. The detailed changes are as follows:

Changes Made:

Main text, Line 186 - 200

Added: The 1D nature of CNTs leads to strongly anisotropic optical absorption [26]. When the polarization of incident light is parallel to the nanotube axis, strong absorption occurs through transitions between subbands with the same index such as the S_{22} transition in semiconducting CNTs at 1.2 eV, the M_{11} transition in metallic CNTs at 1.7 eV, and the DUV response (labeled π) at 4.4 eV. Specifically, the DUV spectral features originate from interband transitions associated with the M point of the graphene Brillouin zone [27, 28], and the transition energies depend on the diameters and atomic structures of CNTs [27, 29, 30]. In contrast, for perpendicular polarization, different interband transitions are excited due to optical selection rules, but their intensities are suppressed because of the depolarization effect [31, 32]. The DUV absorption peak shifts to 4.87 eV for perpendicular polarization. Such a shift is due to the excitation of different interband transitions, while unambiguous assignments of these transitions remain elusive and require systematic studies of aligned CNT films of various diameters and atomic structures [33].

Reviewer #1's comment #3: For the in situ mechanical rotation method, CNTs assemblies appear to have a spiral configuration (Fig 2a inset) that is responsible for its chiroptical activity. The author claimed that the CNTs assemble into a spiral castle (page 9, line 403). I personally find it interesting, but a systematic investigation is missing. I would like the authors to provide further information (e.g., structural characterization) on the assembly formation during the rotational motion and the processing factors affecting this chiral assembly.

Response to comment #3: We thank the reviewer for these valuable questions. First, along the light propagation direction, which is perpendicular to the film, the orientation of aligned CNTs cannot be homogenous. Please also see our response to Reviewer #2's comment #3. Our electromagnetic analyses have shown that a 2D spiral rotation of CNT orientation, such as one illustrated in **Fig. R1C3-1(a)**, is ***not responsible*** for the observed chiroptical response. Instead, to observe CD signals, we need the light to propagate along the axis of a 3D helix; see **Fig. R1C3-1(b)**. Hence, the effective structures created by both mechanical rotation and twist-stacking methods are ***essentially the same, i.e., twisted stacks***; see

Fig. R1C3-1(b). To elaborate on this point, we have further performed full-wave electromagnetic finite-difference time-domain (FDTD) simulations using Ansys Lumerical software. Specifically, we have simulated the two structures shown in **Fig. R1C3-1(c)**. One is a film with 4 rotation angles (0° , 22.5° , 45° , and 67.5°) in 4 quadrants. Each has a film thickness of 40 nm, and there is no angle rotation along the thickness direction. The other film is a twisted stack with the same rotation angles. **Figure R1C3-1(c)** displays CD spectra simulated for these two films, clearly showing that only the 3D helix can induce CD.

Fig. R1C3-1: Illustrations of (a) a 2D spiral pattern and (b) 3D helical twisted stack structures. (c) FDTD simulation CD spectra of films with in-plane and out-of-plane rotation 2D spiral and 3D helical structures.

Hence, the assembly formation during the mechanical-rotation-assisted vacuum filtration process is as follows: CNTs are deposited on the filter membrane to form layers of alignment structures at first, and then alignment orientation between layers is rotated by the flow field created by the mechanical rotation that starts later and for a short period of time (e.g., 30 seconds). For this revision, we have performed a systematic investigation of processing factors of *in situ* mechanical rotation with a new mechanical orbital shaker (Lab Companion OS-2000). Furthermore, we have performed systematic CD mapping characterizations on obtained films.

Specifically, in the new orbital shaking system, we can control not only the rotation speed but also the rotation direction (i.e., clockwise or counterclockwise). Regarding the rotation speed, we have performed experiments at speeds ranging from 50 RPM to 220 RPM for rotating 30 seconds in the middle of conventional vacuum filtration. As shown in **Fig. R1C3-2(a)**, when the rotation speed is low (e.g., 50 RPM), the obtained films are uniform. When the rotation speed increases to 140 RPM, obtained films display spiral patterns because of the formation of *a flow vortex* inside the filtration funnel. This is what we showed in the original manuscript. When the rotation speed further increases (e.g., 220 RPM), the strong flow turbulence created during rotation leads to nonuniform films, which are less useful for practical applications. Hence, we mainly focus on films obtained at low and medium speeds, such as 50 RPM and 140 RPM.

In addition, we can control the rotation direction with the new mechanical shaker (Lab Companion OS-2000), which was not feasible with the mechanical shaker used in the original manuscript (Wincom KJ-201BD). As shown in **Fig. R1C3-2(b)**, at 140 RPM rotation speed, the change of rotation direction leads to opposite spiral patterns.

Fig. R1C3-2: Photos of obtained films through mechanical rotation (a) at different rotation speeds and (b) rotation directions (clockwise and counter-clockwise rotations).

For samples obtained at 140 RPM rotation speed, we have performed detailed CD mapping at different locations of obtained samples. We cut the obtained film into multiple pieces, transferred each on a substrate, and characterized piece by piece with a 2.2-mm diameter incident light spot size. As shown in **Fig. R1C3-3(a)**, we have observed a new phenomenon. Specifically, we have observed different signs of CD at different locations. Based on our prior work (Komatsu *et al.*, “Groove-assisted global spontaneous alignment of carbon nanotubes in vacuum filtration,” *Nano Lett.* 20, 2332-2338 (2020)), there are pre-existing long grooves on filter membranes, which determine the alignment direction of CNTs when they are deposited on filter membranes during vacuum filtration. Along the line crossing the film center that is parallel to the direction of pre-existing grooves on the filter membrane (the line crossing positions (1), (3), (7), (11), (13)), the signs of CD are the same. In contrast, as the line connecting the sample location and the center moves toward the direction perpendicular to these pre-existing grooves, the CD signals become flipped to the opposite sign. The distribution of CD signs along lines parallel and perpendicular to the groove direction indeed provides strong hints on the assembly formation process and indicates that a 3D helical twist structure is the main reason for observed CD signals.

For example, we take positions (3) and (6) in **Fig. R1C3-3(a)** for explanations. In both positions, there are a few layers of aligned CNTs formed near the filter membrane at the beginning of the filtration process. Afterward, the vortex flow field inside the funnel creates a spiral orientation of CNTs on top of it and the local orientations of CNTs at positions (3) and (6) have opposite rotations with respect to the bottom alignment direction, leading to opposite signs of CD. The illustration in **Fig. R1C3-4(a)** shows a schematic of twisted structures. The yellow lines indicate the bottom aligned direction, and there is a spiral pattern (black lines) formed on top. Depending on the relative orientation of the bottom alignment

direction and the local spiral direction, different areas can display different twist handedness (L or R configurations). Hence, with mechanical-rotation-assisted vacuum filtration, we can not only create CNT twist stacks in a ***faster*** manner than the twist-stacking method but also create ***both chirality simultaneously*** in a single sample and a single run. Furthermore, we can flip the sign of CD everywhere in films by changing the rotation direction; see Fig. R2C4 below. Note that the illustration in Fig. R1C3-4(a) displays a simplified two-layer stacking scheme to qualitatively explain the observed CD mapping. In real samples, there could be a continuous twist during mechanical rotation, as illustrated in Fig. R1C3-4(b).

Fig. R1C3-3: (a) LD^f and CD mappings of obtained CNT films prepared using mechanical-rotation-assisted vacuum filtration at 140 RPM. (b) Polarization angle-dependent attenuation for a representative CNT sample, where maximum (A_{\parallel}) and minimum attenuation (A_{\perp}) are used to calculate LD^f.

Fig. R1C3-4: Illustrations of (a) simplified and (b) potentially practical twisted structures formed during mechanical-motion-assisted vacuum filtration.

Fig. R1C3-5: LD^r and CD mappings of obtained CNT films prepared using mechanical-rotation-assisted vacuum filtration at 50 RPM.

If the rotation speed is lower, such as 50 RPM, a vortex flow cannot be formed inside the filtration funnel. Hence, there is no clear spiral pattern formed; see **Fig. R1C3-2(a)**. As a result, within the whole film, the rotation between twisted layers is uniform and the signs of CD signals remain the same, as shown in **Fig. R1C3-5**. This further confirms that for producing any CD signal the light needs to propagate along the axis of a helical structure. Together with CD mapping, we have also used a single-color 633-nm laser diode and polarization optics to perform the mapping of reduced linear dichroism (LD^r) at 1.88 eV in films produced at 140 RPM and 50 RPM, as shown in **Fig. R1C3-3(a)** and **R1C3-5**. **Figure R1C3-3(b)** shows the polarization angle-dependent attenuation of a representative CNT film. LD^r was calculated as $2(A_{//} - A_{\perp}) / (A_{//} + A_{\perp})$, where $A_{//}$ and A_{\perp} are the maximum and minimum attenuation, respectively.

Finally, we would like to make a few remarks on the rotation starting time and time duration. Although in the original manuscript we gave the impression that CD signals are stronger if the rotation

time duration is longer, with further experiments performed for this revision we have discovered that the strength of CD signals is not dependent on the rotation time duration in a monotonic manner. Other factors, including the rotation start time, the solution height, the acceleration rate, and the deceleration rate, also play important roles. The essential point is that the CD signal size depends on how the flow vortex is generated inside the funnel when the funnel is rotated off-axis. Systemic and accurate control of vortex creation, flow movement, and thus the stacking structure of CNTs is a non-trivial task and can be a complex fluid dynamics problem involving many unknown parameters. Thus, to avoid confusion, we have removed any discussion on the rotation time duration dependence of CD signals and acknowledge that future systematic investigation on system fluid dynamics is worthwhile.

To make the logic of the whole manuscript more coherent, **in the revised manuscript, we have flipped the order of introducing twist-stacking and mechanical rotation approaches, added new data in figures, added multiple paragraphs, and added new method sections. In the revised SI, we have added new figures as well. The detailed changes are as follows:**

Changes Made:

Main text, Line 66 – 68, Line 140 – 142

Added: Furthermore, the mechanical rotation method not only accelerated the fabrication of twisted structures but also produced both chiralities simultaneously in a single sample, in a single run, and in a controllable manner.

Main text, Line 370 - 458

Added: Rotational shaking was applied for a short period of time in the middle of the filtration process; see Methods for more details, Supplementary Fig. 9 for an illustration, and Supplementary Video 1 for an experimental demonstration. During the mechanical-motion-assisted CVF process, individual CNTs are first deposited on the filter membrane to form alignment layers. Mechanical rotation starts afterwards and lasts for a certain time duration. While the filtration system is in rotation, a flow field is generated, which controls the alignment orientation of CNTs as CNTs are further deposited to add layers (Fig. 4a). We controlled the rotation speed from 50 to 220 RPM; CNT films fabricated with different morphology are shown in Supplementary Fig. 10. When the rotation speed was high, such as 200 and 220 RPM, strong flow turbulence created inside the funnel led to the formation of nonuniform films, which are less useful for practical applications. When the rotation speed was in an intermediate range of 100-150 RPM, a vortex flow, or a “tornado,” was formed inside the filtration funnel and created a spiral pattern of CNT alignment orientation, as shown in Fig. 4b. Furthermore, the handedness of the tornado was controlled by the direction of mechanical rotation, i.e., clockwise or counterclockwise. The signs of ellipticity around the center of such tornado CNT films produced under opposite mechanical rotation directions were also opposite (Fig. 4c). In addition, CD spectra measured using ellipsometry and four-configuration measurements showed excellent agreement; see Supplementary Fig. 11 and Fig. 12.

We further performed detailed spatial mapping of CD spectra in a tornado CNT film by cutting the film into multiple pieces and characterizing piece by piece with a ~2.2 mm beam size in diameter. Interestingly, we observed different signs of CD at different locations; see Supplementary Fig. 13. Based on our prior work [40], we know that there are pre-existing aligned long grooves on filter membranes, which determine the alignment direction of CNTs when they are deposited on filter membranes during CVF. Along the lines crossing the film center that are parallel and perpendicular to the direction of the pre-existing grooves, the signs of ellipticity were opposite. This is due to the co-existence of left- and right-handed twisted structures formed between macroscopic alignment layers and spiral rotation layers, as illustrated in Supplementary Fig. 14. Note that, fundamentally, light propagation along the axis of a 3D helical structure is the origin of the observed chiroptical responses, which was confirmed through finite-difference-time-domain simulations; see Methods and Supplementary Fig. 15. Hence, the mechanical-

rotation approach can not only create twisted CNT structures faster than the twist-stacking approach but also create both chiralities simultaneously in a single sample, in a single run, and in a controllable manner. In addition to CD mapping, we performed spatial mapping of reduced linear dichroism (LD^r) at 1.88 eV (Methods and Supplementary Fig. 13), as well as mapping of thickness, peak ellipticity, and LD^r at 1.88 eV along the radial direction (Supplementary Fig. 16). Both the thickness and ellipticity were the largest at the center, and thus, center films will be good candidates for applications.

Furthermore, when the mechanical rotation speed was low, such as 50 RPM, the flow vortex was not formed and thus the obtained films were uniform (Supplementary Fig. 10). As a result, the ellipticity sign was constant without any flip within the whole film (Supplementary Fig. 17). This observation further confirms that light propagation along the axis of a 3D helical structure is responsible for chiroptical responses. Finally, although it is challenging to have accurate information on some of the structural parameters (e.g., the thicknesses and angles) of twisted structures formed during mechanical-rotation-assisted CVF, it is theoretically feasible to model obtained films through the effective medium theory by solving wave equations in chiral and bi-isotropic media [41]. The effective parameters describing chiroptical response can be extracted through fitting to experimental data. The *in situ* mechanical-motion-assisted CVF method creates new research opportunities for experimentally controlling and theoretically investigating flow dynamics in the vacuum filtration process to engineer chiroptical response.

Main text

Added: Figures 4b and 4c

Main text, Line 540 – 545

Added: For LD^r mapping, linearly polarized optical absorption was measured using a 660-nm laser diode with a beam diameter of 2.2 mm, polarizers, and half-wave plates. Supplementary Figure. 13b displays a polarization angle-dependent attenuation at 1.88 eV. LD^r was calculated as $2(A_{\parallel} - A_{\perp})/(A_{\parallel} + A_{\perp})$, where A_{\parallel} and A_{\perp} are the maximum and minimum attenuation, respectively.

Main text, Line 837 – 851

Added red texts:

Mechanical-rotation-assisted CVF - The filtration system was mounted on top of an orbital shaker (Lab Companion OS-2000) using a 100 mL flask clamp. The overall system was rotated in a circle with a diameter of 19.1 mm. The rotation speed was controllable in a range of 20 to 500 RPM, and the rotation duration was 30 seconds.

FDTD simulations - The FDTD simulations were done with Ansys Lumerical software. The dielectric function tensor $\boldsymbol{\varepsilon}$ of aligned CNT films was from Fig. 3d. When the alignment orientation was rotated, the dielectric function tensor became

$$\boldsymbol{\varepsilon} \mathbf{R} \mathbf{R}^{-1}, \text{ with } \mathbf{R} = \begin{pmatrix} \cos\theta & -\sin\theta \\ \sin\theta & \cos\theta \end{pmatrix},$$

where θ is the rotation angle. Bloch boundary conditions were used in the sample plane.

SI

Added: Supplementary Figures 10, 13, 14, 15, and 17.

Reviewer #1's comment #4: The authors should provide an AFM height profile to determine the thickness of the spiral CNT assembly per Fig 2. I would assume that this is different from the aligned anisotropic assembly per Fig 1.

Response to comment #4: The thickness of the spiral CNT assembly shown in **Fig. 2** of the original manuscript (**Fig. 4** of the revised manuscript) was nonuniform, and since AFM can only provide local height information (e.g., in a $50 \mu\text{m}^2$ area), it is time-consuming and challenging to have a global evaluation of the thickness profile over the sample. Hence, instead of AFM height profiling, we have performed mapping of the average optical absorption at different positions of samples. The probe light beam had a ~ 2.2 -mm beam size in diameter. Note that the average optical absorption, which is the average of the absorbance values for left- and right-handed circularly polarized light, is independent of the degree of twist inside the sample and is only dependent on the sample thickness (via Beer's law). We benchmarked the average optical absorption for the spiral CNT assembly shown in **Fig. 2** of the original manuscript with respect to the average optical absorption of aligned CNTs in **Fig. 1** of the original manuscript. Similarly, the average optical absorption of aligned CNTs, which is the average absorbance between the parallel and perpendicular polarization configurations, is independent of the degree of alignment inside the sample and is only dependent on the sample thickness. **Figure R1C4(a)** displays a photo of a film obtained at 140 RPM rotation, which shows a spiral pattern. **Figure R1C4(b)** summarizes thickness mapping at different locations of the obtained sample. The thickness distribution of films produced using mechanical-rotation-assisted vacuum filtration varied little from sample to sample. In addition, we have performed radial mapping of the peak ellipticity and LD^r at 1.88 eV shown in **Fig. R1C4(c)** and **Fig. R1C4(d)** at the same positions as shown in **Fig. R1C4(a)**. Both thickness and ellipticity were the largest at the center, and thus, center films will be good candidates for applications.

Fig. R1C4: (a) Photos, (b) thickness mapping, (c) CD mapping, and (d) LD mapping of obtained CNT films prepared using mechanical-rotation-assisted vacuum filtration at 140 RPM.

In the revised manuscript, we have added a few new sentences to address the above points. In the revised SI, we have added the data of thickness mapping as new figures. The detailed changes are as follows:

Changes Made:

Main text, Line 409 – 412, 441 – 442

Added: In addition to CD mapping, we performed spatial mapping of reduced linear dichroism (LD^r) at 1.88 eV (Methods and Supplementary Fig. 13), as well as mapping of thickness, peak ellipticity, and LD^r at 1.88 eV along the radial direction (Supplementary Fig. 16). Both the thickness and ellipticity were the largest at the center, and thus, center films will be good candidates for applications.

SI

Added: Supplementary Figure 16.

Reviewer #1's comment #5: For the *in situ* mechanical rotation method, quantitative processing-structure-chiral property relationships should be provided. For instance, what is different for the assembly after 70s-rotation that leads to a large ellipticity value? What is the effect of the strength of the rotational force on the chiroptical response of the assembly?

Response to comment #5: This comment is related to the reviewer's comment #3. As mentioned before, first, light propagation along the axis of a helical structure leads to CD signals. Hence, the CD signal is a good quantitative measure of the degree of twist (i.e., structural property information) inside obtained samples. We have tried to utilize scanning electron microscopy (SEM) to characterize twist structures. However, such characterization requires a *destructive* etching process to destroy the sample layer by layer, which makes it inappropriate for quantitative characterization.

Second, as we mentioned in our response to comment #3, we have performed systematic experiments on the *in situ* rotation method and systematic characterization studies on obtained films. We interpret the 'rotational force' mentioned by the reviewer as the centrifugal force, which is controlled by the rotation speed. Specifically, we performed experiments ranging from 50 RPM to 220 RPM for rotating in a short period of time (30 seconds) in the middle of conventional vacuum filtration. As shown in **Fig. R1C3-2(a)**, when the rotation speed was low (e.g., 50 RPM), the obtained films were uniform. When the rotation speed increased to 140 RPM, obtained films displayed spiral patterns, as described in the original manuscript. When the rotation speed further increased (e.g., 220 RPM), the strong turbulence created during rotation led to nonuniform films. CD mapping of obtained films at 140 RPM showed different signs of CD at different locations, while obtained films at 50 RPM displayed uniform CD signs across films; see **Fig. R1C3-3(a)** and **R1C3-5**. These observations not only reveal structure-chiral property relationship but also are *consistent* with the twist-stack picture. Furthermore, the strength of CD signals in films obtained at 50 RPM was generally small, because of the small rotation force applied on CNTs to form twist structures.

Finally, as we mentioned in our response to comment #3, we envision that fluid dynamics in the funnel of a revolving filtration system are crucial to the chiral properties of obtained films, which can complicatedly depend on the rotation time duration, when rotation starts, solution height, acceleration, and deceleration rates. Thus, to avoid confusion, we have removed any discussion on the rotation time duration dependence of CD signals and acknowledge that future systematic investigation on system fluid dynamics is worthwhile.

The detailed changes are the same as those in response to this reviewer's comment #3.

Reviewer #1's comment #6: Artificial twisted stacking as a strategy has been applied to various nanostructures to elicit artificial chirality, beyond the chiral bilayer graphene case the authors mentioned (Kim et al. Nat. Nanotechnol. 2016, 11, 520), i.e., the method itself is not conceptually novel. Some more exemplary systems and recent works/reviews are missing such as (1) Han et al. Adv. Mater. 2023, 35, 2206141; (2) Lv et al. Angew. Chem. Int. Ed. 2017, 56, 5055–5060; (3) Wu et al. Nano Lett. 2021, 21, 8298–8303; etc. The authors should better describe these and balance their references accordingly.

Response to comment #6: We thank the reviewer for these references. We agree that twisted structures make one of the most versatile approaches/platforms to create structure-induced chirality in many

contexts, such as twisted metasurfaces, nanowires, and 2D materials, as summarized in “*Han et al., Adv. Mater., 35, 2206141 (2023)*”. For example, as specifically mentioned by the reviewer, two other papers, “*Lv et al. Angew. Chem. Int. Ed. 56, 5055 – 5060 (2017)*” and “*Wu et al. Nano Lett., 21, 8298-8303 (2021)*,” demonstrated CD response in twisted gold nanowires. However, our work has the following features that distinctly differentiate ours from the previous studies:

First, in most previous demonstrations including the three papers mentioned by the reviewer, the observed CD signals were comprised of both *real* CD signals from structure-induced chirality and artificial CD signals coming from linear dichroism (LD) and linear birefringence (LB) effects. Without removing the linear-optical-anisotropy-induced CD response, it is hard to distinguish whether artificial structures created the structure-induced chirality. For example, in our experiments (e.g., **Fig. 2b** in the original manuscript and **Supplementary Fig. 3** in the revised manuscript), even with perfectly aligned CNTs without structure-induced chirality, one CD measurement under one configuration can produce substantial CD signals. Hence, it is crucial to perform careful CD spectroscopy measurements on solid-state samples with anisotropy. Our four-configuration measurement protocol removes such linear optical anisotropy effects, and their results show excellent agreement with ellipsometry spectroscopy characterization, which is free from linear optical anisotropy effects.

Second, as we mentioned in our response to comment #1, CNTs are 1D *quantum* nanostructures and their optical properties are governed by quantum confinement effects. This is in stark contrast to other nanowires (e.g., gold nanowires), whose properties in all directions are still determined by bulk materials. For gold nanowires, the optical anisotropy originates from classical plasmonic resonances, and twisted stacks of these nanowires can produce CD signals. Our twisted stacks of aligned CNTs offer the first demonstration of synthetic chirality from wafer-scale assemblies of 1D *quantum* nanostructures.

Third, the material platforms that can produce CD responses in the DUV range are quite limited. Especially for metamaterials created through top-down methods, DUV operation requires sophisticated nanofabrication processes. Thanks to the quantum-confinement-enhanced strong DUV response of CNTs, the observed DUV CD signals of CNT twisted stacks are the strongest to date.

In the main text, we have revised the introduction and concluding paragraphs and added a few sentences in the main body to further highlight the novelty of our material platform and characterization technique. We have also added new references mentioned by the reviewer. The detailed changes are as follows:

Changes Made:

Main text, Line 102 - 120

Added red texts: Natural molecules generally have weak chiroptical responses because of their small sizes compared to the light wavelength. Chiral metamaterials, consisting of periodic artificial symmetry-breaking structures such as twisted structures manufactured with either top-down or bottom-up methods, can boost chiroptical responses through resonance enhancement [11, 12]. However, top-down manufacturing requires sophisticated nanofabrication facilities and intricate processes to create engineered structures [13], especially for short-wavelength applications that necessitate ultrasmall feature sizes. Bottom-up assembling of one-dimensional (1D) objects offers a simple and scalable manufacturing route, while current demonstrations are constrained to simple metallic or dielectric rods with limited conventional material physical properties [14 - 18]. Nanomaterials with room-temperature quantum-confinement effects and their artificial architectures have recently emerged as new chiral platforms. For example, chirally stacked multiple layers of graphene have displayed circular dichroism (CD), which is defined as the differential absorption of left and right circularly polarized light [19]. New methods that can create artificial matter based on such nanomaterials with strong and controllable chirality without involving complicated procedures are being sought.

Main text, Line 268 - 270

Added: Note that such elimination of CD signals induced by linear optical anisotropy was not performed in a large majority of previous studies of chiral metamaterials and nanomaterials.

Main text, Line 482 - 490

Added red texts: Rather, CNTs are quantum wires with extreme quantum confinement, leading to quantized energy levels, or subbands, with energy separation (~ 1 eV) much larger than the room-temperature thermal energy (26 meV). Optical transitions between such 1D subbands with van Hove singularities in the joint density of states in CNTs manifest themselves as peaks in room-temperature optical spectra. The fact that our observed wavelength-dependent giant CD signals are due to these quantum-confinement-induced optical properties enables the engineering of CD signals through the quantum engineering of CNT diameters and atomic structures.

Response to Reviewer #2

Reviewer #2's comment #1: Although the alignment technique has already been published before (ref 13 and 14 in the manuscript), I suggest that the authors give a short summary of the mechanism that leads to the orientation of the CNT over such large areas.

Response to comment #1: In the revised manuscript, we have added a short summary of the alignment mechanism occurring during vacuum filtration. The detailed changes are as follows:

Changes Made:

Main text, Line 170 – 173

Added: As CNTs accumulate near the surface of the filter membrane during the filtration process, a liquid-crystal phase transition leads to wafer-scale alignment of CNTs [20, 21]. The CVF process is spontaneous and does not require any external stimulus.

Reviewer #2's comment #2: There are many other examples of similar structures of twisted layers of oriented 1D nano-objects. Twisted layers of CNT have for instance already been prepared for their enhanced mechanical properties (DOI: 10.1002/adma.201605750). The chiroptical properties of Bouligand nanostructures was also widely studied. This structure is for instance found to occur naturally in the cuticle of some beetles (doi.org/10.1364/AO.49.004558) or in films of cellulose nanocrystals (DOI: 10.1002/anie.201500277), and has been synthetically prepared with metallic oxide nanowires (DOI: 10.1002/anie.201903264) or metallic nanowires/nanorods (see for instance doi.org/10.1021/acsnano.1c04804, 10.1021/acs.nanolett.1c02812 or DOI: 10.1038/ncomms1877). This should be better highlighted in the introduction, and the optical properties of the CNT films should be compared to the other systems.

Response to comment #1: We thank the reviewer for these references. This comment is also relevant to Reviewer #1's comment #6; please also refer to our response there, especially for a comparison of our work with prior works. Among the references mentioned by the reviewer, the paper DOI: 10.1002/adma.201605750 describes a 3D printing approach to fabricate Bouligand or twisted plywood structures of polymer/multiwall CNT composites. The alignment of multiwall CNTs was created by applying a huge electric voltage (900 V) during 3D printing, and the alignment direction was rotated through a rotation stage. This paper only discussed mechanical properties. In comparison with our approach, the approach used in this paper had several disadvantages. First, large electric voltages were needed for alignment, while, in our approach, CNTs align *spontaneously*. Second, 3D printing requires polymer resins so that the resultant structures in the reference paper were not purely CNTs. Optical properties can be limited by the existence of polymers; for example, polymers can introduce additional strong UV absorption. Third, our manual twist-stacking approach is versatile enough to incorporate other materials and create a large variety of multilayer structures for engineering chiroptic response, which would be challenging to achieve using the reported approach in the reference paper.

We agree with the reviewer that the chiroptic responses of such Bouligand or twisted structures have been studied in a variety of material systems, such as in the other papers mentioned by the reviewer (doi.org/10.1364/AO.49.004558, 10.1002/anie.201903264, doi.org/10.1021/acsnano.1c04804, 10.1021/acs.nanolett.1c02812, 10.1038/ncomms1877). However, the optical properties of all reported materials are still governed by bulk material properties, whereas the optical properties of CNTs are governed by quantum-confinement effects. For example, CNT DUV optical absorption features are characteristic of interband transitions near the M point of the hexagonal Brillouin zone of graphene, which are also dominated by CNT quantum confinement. The diameter- and atomic structure-dependent

DUV spectral features have been theoretically and experimentally demonstrated. Our work opens new opportunities of creating and engineering chiral matter based on 1D quantum nanostructures (not limited to CNTs) for exploring novel chiral photonic and optoelectronic devices and beyond.

In the main text, we have revised the introduction and concluding paragraphs and added a few sentences in the main body to further highlight the difference and novelty of our material platform and fabrication method. We have also added new references mentioned by the reviewer. The detailed changes are as follows:

Changes Made:

Main text, Line 102 - 120

Added red texts: Natural molecules generally have weak chiroptical responses because of their small sizes compared to the light wavelength. Chiral metamaterials, consisting of periodic artificial symmetry-breaking structures such as twisted structures manufactured with either top-down or bottom-up methods, can boost chiroptical responses through resonance enhancement [11, 12]. However, top-down manufacturing requires sophisticated nanofabrication facilities and intricate processes to create engineered structures [13], especially for short-wavelength applications that necessitate ultrasmall feature sizes. Bottom-up assembling of one-dimensional (1D) objects offers a simple and scalable manufacturing route, while current demonstrations are constrained to simple metallic or dielectric rods with limited conventional material physical properties [14 - 18]. Nanomaterials with room-temperature quantum-confinement effects and their artificial architectures have recently emerged as new chiral platforms. For example, chirally stacked multiple layers of graphene have displayed circular dichroism (CD), which is defined as the differential absorption of left and right circularly polarized light [19]. New methods that can create artificial matter based on such nanomaterials with strong and controllable chirality without involving complicated procedures are being sought.

Main text, Line 210 – 212

Added: Compared to other methods of creating twisted structures using huge electric voltage and polymer/CNT composites [34], our method is simpler and more scalable and can produce CNT-only structures.

Main text, Line 482 – 490

Added red texts: Rather, CNTs are quantum wires with extreme quantum confinement, leading to quantized energy levels, or subbands, with energy separation (~ 1 eV) much larger than the room-temperature thermal energy (26 meV). Optical transitions between such 1D subbands with van Hove singularities in the joint density of states in CNTs manifest themselves as peaks in room-temperature optical spectra. The fact that our observed wavelength-dependent giant CD signals are due to these quantum-confinement-induced optical properties enables the engineering of CD signals through the quantum engineering of CNT diameters and atomic structures.

Reviewer #2's comment #3: The sample obtained with mechanical-rotation-assisted CVF shows a spiral-like pattern on the substrate. Is the CD measurement done in the center of this spiral or away from the center? Is the CD spectrum dependent on the position at which the measurement is done? I think that the RC2 Mueller ellipsometer that was used can do mapping of the sample, I suggest that the authors add such a large-area mapping of the CD over the substrate to probe the homogeneity of the chiroptical properties. As discussed in the text, linear dichroic effects are quite large in such samples. It would be good to show a mapping of the LD over the sample along with CD mapping. Furthermore, is the local orientation of the CNT homogeneous along the thickness of the film (parallel stacking of CNT when looking perpendicular to the film)?

Response to comment #3: We thank the reviewer for this constructive comment. This comment is also related to Reviewer #1's comment #3 and comment #5. Please also refer to our responses to these two comments. Most of the CD measurements presented in the original manuscript were taken for films around the center. For this revision, we have performed systematic CD mapping on films fabricated through mechanical-rotation-assisted vacuum filtration, instead of characterizing only the center, and discovered that CD signals are strongly position-dependent.

As we described in the original manuscript, the four-configuration approach with a conventional CD spectrometer can have the same results as an RC2 ellipsometer. Hence, for the revision, we have performed CD mapping by cutting fabricated films into pieces, transferring each on a substrate, and measuring CD spectra piece by piece using the four-configuration approach. The beam diameter of the collimated incident beam for both the four-configuration and ellipsometry measurements was 2.2 mm. The reviewer may be referring to the mapping capability with a much higher spatial resolution (e.g., μm^2) of the RC2 ellipsometer. To achieve such a resolution, focusing probes need to be added. However, it is **not** possible with the employed standard vertical base RC2 models, and hence, specialized customization will be required. Furthermore, addition of focusing probes can cause non-ideality in the Mueller matrix data due to the so-called delta offset from lenses. Our current measurements with collimated beams can provide the purest Mueller matrix data. Furthermore, we would like to emphasize that linear dichroic effects, including linear dichroism (LD) and linear birefringence (LB) effects, have been **removed** in obtained CD signals using both the four-configuration approach and ellipsometry.

The obtained CD mapping is shown in **Fig. R1C3-3(a)**, which shows that the CD sign and size are dependent on the sample position. Such dependence originates from the relative orientation of the bottom aligned layers, which is formed due to pre-existing grooves on filter membranes, and the local spiral pattern direction, which is formed due to mechanical rotation. Please refer to our response to Reviewer #1's comment #3 for more details and **Fig. R1C3-4(a)** for the illustration. Hence, mechanical-rotation-assisted vacuum filtration can not only create CNT twist stacks in a **faster** manner than the twist-stacking method but also create **both chirality simultaneously** in a single sample and in a single run.

Regarding the reviewer's final question, we would like to point out that if the local orientation of aligned CNTs is homogenous perpendicular to the film, there should be no CD signals. We have proven this through newly performed full-wave electromagnetic FDTD simulations using Ansys Lumerical software. Please also refer to our response to Reviewer #1's comment #3 for more details and **Fig. R1C3-1(c)** for new simulation results, which clearly show that only when the light propagates along the axis of a 3D helical structure (i.e., twisted stacks), CD can exist. Hence, when looking perpendicular to the film, the local orientation of CNTs must be rotated and not homogenous along the film thickness direction.

Finally, linear dichroic effects have been removed from all obtained CD spectra using the four-configuration method. We have utilized a 633-nm laser diode and corresponding polarization optics to perform the mapping of reduced LD (LD^r) at 1.88 eV, as shown in **Fig. R1C3-3**.

To make the logic of the whole manuscript more coherent, **in the revised manuscript, we have flipped the order of introducing twist-stacking and mechanical rotation approaches, added new data in figures, added multiple paragraphs, and added new method sections. In the revised SI, we have added new figures as well. The detailed changes are as follows:**

Changes Made:

Main text, Line 392 – 412, 441 – 442

Added: We further performed detailed spatial mapping of CD spectra in a tornado CNT film by cutting the film into multiple pieces and characterizing piece by piece with a ~2.2 mm beam size in diameter. Interestingly, we observed different signs of CD at different locations; see Supplementary Fig. 13. Based on our prior work [40], we know that there are pre-existing aligned long grooves on filter membranes, which determine the alignment direction of CNTs when they are deposited on filter membranes during CVF. Along the lines crossing the film center that are parallel and perpendicular to the direction of the pre-existing grooves, the signs of ellipticity were opposite. This is due to the co-existence of left- and

right-handed twisted structures formed between macroscopic alignment layers and spiral rotation layers, as illustrated in Supplementary Fig. 14. Note that, fundamentally, light propagation along the axis of a 3D helical structure is the origin of the observed chiroptical responses, which was confirmed through finite-difference-time-domain simulations; see Methods and Supplementary Fig. 15. Hence, the mechanical-rotation approach can not only create twisted CNT structures faster than the twist-stacking approach but also create both chiralities simultaneously in a single sample, in a single run, and in a controllable manner. In addition to CD mapping, we performed spatial mapping of reduced linear dichroism (LD^r) at 1.88 eV (Methods and Supplementary Fig. 13), as well as mapping of thickness, peak ellipticity, and LD^r at 1.88 eV along the radial direction (Supplementary Fig. 16). Both the thickness and ellipticity were the largest at the center, and thus, center films will be good candidates for applications.

Main text, Line 837 – 851

Added red texts:

FDTD simulations - The FDTD simulations were done with Ansys Lumerical software. The dielectric function tensor ϵ of aligned CNT films was from Fig. 3d. When the alignment orientation was rotated, the dielectric function tensor became

$$R\epsilon R^{-1}, \text{ with } R = \begin{pmatrix} \cos\theta & -\sin\theta \\ \sin\theta & \cos\theta \end{pmatrix},$$

where θ is the rotation angle. Bloch boundary conditions were used in the sample plane.

SI

Added: Supplementary Figures 10, 13, 14, and 15.

Reviewer #2's comment #4: Fig. 2c and Fig. 2d show reversed CD spectra. Were those measurements done on samples with a different handedness, or the sample was simply reversed (as shown in Fig. 2b). This information should be added in the text or in the caption. Furthermore, is the CD spectrum reversed when the orbital shaker is rotated in the opposite direction?

Response to comment #4: We greatly appreciate this illuminating comment. To address this point, we further examined our data and performed many additional measurements. Please note that this comment is also closely related to comment #3 above. Both data shown in **Fig. 2c** and **Fig. 2d** of the original manuscript are results of taking the average of CD spectra under four configurations. This means that the samples shown in **Fig. 2c** and **Fig. 2d** of the original manuscript have opposite handedness.

In our new experiments of CD mapping, we have found that, even within the same sample, different positions can have CD signals with opposite signs and different handedness, which were reproducible and controllable. Detailed explanations were provided in our response to comment #3.

Furthermore, our new experiments employed a new mechanical orbital shaker (Lab Companion OS-2000). In this new model, we can control the rotation direction (i.e., clockwise and counterclockwise), which was not feasible with the mechanical shaker (Wincom KJ-201BD) used for taking the data included in the original manuscript. As shown in **Fig. R2C4(a)**, the change of rotation direction leads to opposite spiral patterns. As a result, CD mapping showed opposite signs at each mapping position. **Fig. R2C4(b)** shows CD spectra around the center area.

Fig. R2C4: (a) Images and (b) CD spectra of films obtained using mechanical rotation at two opposite directions.

In the revised manuscript, we have added new data in Figure 4 and added a few sentences. The detailed changes are as follows:

Changes Made:

Main text, Line 382 - 389

Added: When the rotation speed was in an intermediate range of 100-150 RPM, a vortex flow, or a “tornado,” was formed inside the filtration funnel and created a spiral pattern of CNT alignment orientation, as shown in Fig. 4b. Furthermore, the handedness of the tornado was controlled by the direction of mechanical rotation, i.e., clockwise or counterclockwise. The signs of ellipticity around the center of such tornado CNT films produced under opposite mechanical rotation directions were also opposite (Fig. 4c).

Main text

Added: Figures 4b and 4c

Reviewer #2’s comment #5: A lot of studies rely on the g-factor ($g=\Delta\varepsilon/\varepsilon$) to compare the intrinsic chiroptical properties of different samples. Authors have made the choice to use the ellipticity normalized to the thickness, which also makes sense. It would however be worthwhile to also show (at least) one g-factor spectrum and compare to the values typically obtained in the literature. Indeed, depending on the targeted application, it is important to know if the structure is lossy (absorption) or if the CD arises in a transparent film based only on differences in the refractive index for LCP and RCP light (phase-retardation effect).

Response to comment #5: We agree with the reviewer that it is worthwhile to show one g-factor spectrum. Hence, in the revised SI, we have added a new figure (Supplementary Figure 6) showing a g-factor spectrum for the 3-layer twist-stacking sample with the largest experimentally observed normalized ellipticity (R3 sample in Fig. 2 of the revised manuscript).

We also agree with the reviewer that it is worthwhile to compare the g value of our CNT samples with the reported values in other platforms. Hence, in Supplementary Table 1, we have added a new column for the g factors of different materials and metamaterial platforms in the deep ultraviolet range. We have found that our experimentally measured g factor (0.07) of CNT films produced using the twist-stacking method is already among the platforms that have some of the largest g factors. Note that most natural molecules have g factors in a range of 10^{-4} to 10^{-3} . Furthermore, our theoretical calculations

have demonstrated that normalized ellipticity can be further increased with an increasing number of stacking layers. The largest expected g value for such a system is 0.22. **In the revised manuscript, we have also added some text describing the g factor obtained from theoretical calculations.**

Finally, we would like to clarify that the refractive index difference between LCP and RCP light in transparent films induces the circular birefringence (CB) effect. In contrast, the CD, by definition, is from the absorption difference between LCP and RCP light. Specifically, the extracted anisotropic dielectric constants of aligned CNTs are substantial both in the real and imaginary parts along the tube axis, suggesting that both CD and CB responses exist. In fact, we have identified both CD and CB responses in Mueller matrix analysis of spectroscopic ellipsometry data (**Supplementary Figures 4 and 11 of the revised SI**). **In the revised manuscript, we have added a new sentence.**

The detailed changes are as follows:

Changes Made:

Main text, Line 65 and Line 139

Added red texts: ... will exhibit an ellipticity as high as 150 mdeg/nm, **which corresponds to a g factor of 0.22.**

Main text, Line 267 – 268

Added red texts: **In addition to CD, circular birefringence is also noticeable.**

Main text, Line 305 – 312

Added red texts: **In addition, Supplementary Figure 6 displays the corresponding g-factor spectrum of the three-layer right-handed stack, where the g factor was calculated as the ratio of the differential absorption (i.e., CD) to the average absorption of left and right circularly polarized light. The largest g factor is ~0.07, which is larger than those of natural molecules by 1 to 2 orders of magnitude and is comparable to those of other existing platforms with large g factors; see Supplementary Table 1 for comparison with other chiral platforms and structures.**

SI

Added: **Supplementary Figure 6**

Reviewer #2's comment #6: The modeling by the Transfer Matrix approach is only shown for samples prepared by the twist-stacking of multiple aligned CNT films. Is it also possible to model the samples obtained by mechanical-rotation-assisted CVF?

Response to comment #6: To use the transfer matrix approach, we need to know the structural information of the sample, such as the rotation angles for each layer and layer thickness. However, compared to films fabricated through the deterministic manual twist-stacking method, it is challenging to *precisely* know how twist structures are formed in samples fabricated using the mechanical-rotation-assisted CVF method. Hence, it is currently not possible to directly apply the current transfer matrix method to model samples obtained by mechanical-rotation-assisted CVF.

Instead, it is possible to model the CD response of spiral films in an effective-medium approach. In the general case, the electromagnetic constitutive equations of chiral media can be written as

$$\begin{aligned}\vec{D} &= \boldsymbol{\varepsilon} \cdot \vec{E} + \boldsymbol{\alpha} \cdot \vec{H}, \\ \vec{B} &= \boldsymbol{\beta} \cdot \vec{E} + \boldsymbol{\mu} \cdot \vec{H},\end{aligned}$$

where \vec{D} , \vec{E} , \vec{B} , \vec{H} are displacement field, electric field, magnetic field, and magnetizing field vectors, respectively, $\boldsymbol{\varepsilon}$ and $\boldsymbol{\mu}$ describe permittivity and permeability, and $\boldsymbol{\alpha}$ and $\boldsymbol{\beta}$ describe optical activity. Considering the co-existence of LD in samples, $\boldsymbol{\varepsilon}$, $\boldsymbol{\mu}$, $\boldsymbol{\alpha}$, and $\boldsymbol{\beta}$ are all dyadics. With new relations, the

propagation wave equations in such effective *uniaxial chiral* media can be solved, and the transmission properties of multilayer structures can in principle be calculated through the transfer matrix method as well. However, in this case, the wave equations and transfer matrix become much more complex because fundamental eigenmodes become general elliptically polarized modes. More details can be found in the book: “Lindell, Ismo, et al. *Electromagnetic waves in chiral and bi-isotropic media*. Artech House, 1994.” Once such a framework is fully established (which is beyond the scope of current work), we can fit ϵ , μ , α , and β to match experimentally measured CD spectra and other linearly polarized spectra. The fit quantities can represent the optical anisotropy and chirality in specific samples.

In the revised manuscript, we have added a few new sentences to discuss the effective medium approach. The detailed changes are as follows:

Changes Made:

Main text, Line 448 - 454

Added: Finally, although it is challenging to have accurate information on some of the structural parameters (e.g., the thicknesses and angles) of twisted structures formed during mechanical-rotation-assisted CVF, it is theoretically feasible to model obtained films through the effective medium theory by solving wave equations in chiral and bi-isotropic media [41]. The effective parameters describing chiroptical response can be extracted through fitting to experimental data.

Reviewer #2’s comment #7: It is shown that the CD is maximal for 3 layers oriented at 30°. What happens if 3 layers are oriented with a twist angle of 60°? This would be equivalent to interchange the second and third layer in a 3x30° sample, and I wonder if the CD spectrum is modified by doing so. Furthermore, what happens when more than 3 layers are stacked with a twist angle of 30°. In other words, is the maximal CD obtained for a half turn of the “helix” or does the CD increases further when more layers are added?

Response to comment #7: Figure R2C7-1(a) shows measured CD spectra for two 3-layer stacks at twist angles of 60° and 30°. We unfortunately fail to understand the reviewer’s comment that a 3-layer stack with a twist angle of 60° is equivalent to a 3-layer stack with a twist angle of 30° with interchanged 2nd and 3rd layers. Figure R2C7-1(b) illustrates three configurations of 3-layer stacks, which are a 3-layer stack with a twist angle of 30°, a 3-layer stack with a twist angle of 60°, and a 3-layer stack with a twist angle of 30° with interchanged 2nd and 3rd layers, from the left to right. The middle and the right illustrations have the same configuration of the first two layers but a different last layer (3rd layer).

Fig. R2C7-1: (a) CD spectra of a 3-layer twist stack at a twist angle of 60°. (b) Illustrations of different configurations of 3-layer stacks.

Furthermore, we have performed new transfer matrix simulations on CD spectra for samples with an increasing number of stacking layers at a fixed twist angle of 30° . As shown in **Fig. R2C7-2(a)**, the ellipticity in the unit of mdeg has an overall increasing trend with increasing number of layers but exhibits some oscillations. In contrast, the normalized ellipticity in the unit of mdeg/nm shown in **Fig. R2C7-2(b)** has an overall decreasing trend with increasing number of layers, also exhibiting oscillations. This is because the increasing CD cannot catch up with linearly increasing sample thickness. The oscillation highlights the effects of interlayer reflection and interference.

In the original manuscript, we presented the optimal twist angle that can yield the maximum ellipticity in the unit of mdeg/nm for varying number of layers, calculated using the transfer matrix method. In **Fig. R2C7-2(c)**, in addition to transfer matrix results, we have added a curve from the analytical half-turn-helix equation, which indicates that the optimal angle occurs at $\pi/2N$ with N being the number of stacking layers. Transfer matrix simulations agree with the analytical half-turn-helix calculations well when the number of layers is small. The deviation when the number of layers increases is due to interlayer reflection and interference effects in multiple layers, which is captured in transfer matrix simulations but is not considered in the simple analytical calculation (i.e., the half-turn-helix equation).

Fig. R2C7-2: (a) Ellipticity in the unit of mdeg and (b) normalized ellipticity with respect to thickness in the unit of mdeg/nm as a function of the number of stacked layers. (c) The optimal twist angle as a function of the number of stacked layers obtained using transfer matrix simulations and the half-turn-helix analytical calculation.

In the revised SI, we have added new simulation data in Supplementary Figure 8. We have also added a few sentences in the revised manuscript. The detailed changes are as follows:

Changes Made:

Main text, Line 356 - 363

Added: In addition to the angles obtained from the transfer matrix method, the optimal angles were also calculated using the analytical half-turn-helix equation $\pi/2N$. Transfer matrix results agree well with those of analytical calculations when N is small, while a deviation between the two becomes noticeable when N increases. This deviation is due to interlayer reflection and interference effects in multiple layers, which are captured in the transfer matrix method but are not considered in the simple analytical calculations.

Reviewer #2's comment #8: At the end of the conclusion (p. 9 l. 402), authors mention that “these structures can potentially reveal new phenomena, including non-optical phenomena...”. It should be specified which non-optical phenomena could be revealed.

Response to comment #8: One example of non-optical phenomena is the chirality-induced spin selection (CISS) effect; for a review, see, e.g., Yang, See-Hun *et al.*, “Chiral spintronics,” *Nature Reviews Physics*

3, 328-343 (2021). In the CISS effect, right-handed spins travel through right-handed chiral molecules more easily than left-handed spins, and vice versa. Consequently, the incoming spins become polarized depending on the handedness of chiral molecules such that the transmission through the molecule is spin selective. Specifically, in twisted stacks of aligned CNT films, it is possible that the electron transport along the stacking direction can display spin filtering phenomena.

In the revised manuscript, we have added new phrases in introduction and concluding paragraph and new reference to specify the non-optical phenomena. The detailed changes are as follows:

Changes Made:

Main text, Line 146 – 147

Added red texts: This CNT-based chiral platform will open up new opportunities not only for next-generation photonic and optoelectronic devices, such as chiral (quantum) optical emitters [22], chiral optical sensors, and chiral photodetectors, but also for exploration of new phenomena, including non-optical effects **such as spintronics [23]**, for broader applications.

Main text, Line 494

Added red texts: These structures can potentially reveal new phenomena, including non-optical phenomena **such as spintronics**, and open up new opportunities for next-generation electronic, photonic, and optoelectronic devices.

Response to Reviewer #3

Reviewer #3's comment #1: I notice that there is a mixture of energy and wavelengths in the text. In some cases both are reported, in some other only one. Please make your own choice.

Response to comment #1: We thank the reviewer for this valuable comment. **In the revised manuscript and SI, we have chosen to mention only energy in the text. However, whenever applicable, we have updated all figures to have a secondary x-axis for showing the corresponding wavelength; see our response to the next comment as well.**

Reviewer #3's comment #2: It would be very useful to have both indicated in the horizontal scale (for example eV at bottom and wl at the top) of the plots. Especially in the legend of figure 2, it is not really clear that the ellipticity is normalized to the film thickness.

Response to comment #2: We agree with the reviewer that it is helpful to have both energy and wavelength indicated in bottom and top x axes. **In the revised manuscript and SI, whenever applicable, we have updated all figures to have both energy and wavelength horizontal axes.** In addition, as already mentioned in the original manuscript (now in Line 181 – 183 of the revised manuscript), the ellipticity is normalized by the film thickness for comparing different film samples. **In the revised manuscript and SI, whenever applicable, we have updated all figures in the manuscript and SI to express CD signals in terms of normalized ellipticity.** Furthermore, we feel it may not be clear to readers how CD spectra obtained from ellipsometry are converted to ellipticity in the units of mdeg and mdeg/nm. **Therefore, in the revised manuscript, we have added a few sentences to explain the conversion procedure in the Methods section. The detailed changes are as follows:**

Changes Made:

Main text, Line 628 - 633

Added: The raw CD signal, obtained as the absorbance difference $\Delta A = A_L - A_R$, was converted into the ellipticity (mdeg) = $\tan^{-1} \left\{ \left(I_L^{1/2} - I_R^{1/2} \right) / \left(I_L^{1/2} + I_R^{1/2} \right) \right\} \approx \Delta A / 32980$ (small angle approximation) [25], where $I = I_0 e^{-A \ln 10}$ is the transmitted intensity (Beer's law) and I_0 is the incident intensity of light. Furthermore, we normalized the ellipticity by the sample thickness to express it in mdeg/nm.

Response to Reviewer #4

Reviewer #4's comment #1: In method i), very good agreement of CD spectra determined by standard CD spectroscopy and differential decomposition of the Mueller matrix is reported (figure 2c). However, the authors should discuss the effects of sample inhomogeneity, which is clearly noticed in figure 2a. Sample homogeneity is critical for all possible applications referred in the text.

Response to comment #1: We thank the reviewer for this constructive comment. This comment is also related to Reviewer #1's comment #4. Please refer to our responses there for more details. Briefly, for the revision, we have performed film-thickness mapping by measuring the average optical absorption at different sample positions, which is independent of the degree of polarization and only dependent on sample thickness. The probe light beam had a ~ 2.2 mm beam size in diameter. **Figure R1C4(b)** summarizes the thickness mapping results for a sample with a spiral pattern. The thickness distribution of films produced using mechanical-rotation-assisted vacuum filtration is reproducible and varies little from sample to sample. In addition, we have performed radial mappings of the peak ellipticity (**Fig. R1C4(c)**). Both thickness and ellipticity are the largest at the center, and thus, center films will be good candidates for applications.

The detailed changes are the same as those in response to Reviewer #1's comment #4.

Reviewer #4's comment #2: Do data in figure 2c are from the center of the sample?

Response to comment #2: Most data presented in the original manuscript were from the center. However, for the revision, we have performed systematic CD mapping on films fabricated by mechanical-rotation-assisted vacuum filtration and discovered that CD signals are strongly dependent on the position of measurements. Please refer to our response to Reviewer #1's comment #3 and the corresponding changes we have made.

Reviewer #4's comment #3: How LD, LD', LB, LB', CB, and CD look along a radius?

Response to comment #3: This comment is relevant to Reviewer #1's comment #3, Reviewer #1's comment #4, and Reviewer #2's comment #3. Please refer to our responses to these three comments for more details. For the revision, we have performed additional radial mappings of CD using "four-configuration" CD measurements with a 2.2-mm collimated beam spot size. We anticipate that LD', LB, and LB' have similar trends with LD, and CB has a similar trend to CD. Please refer to **Fig. R1C3-3** (CD and reduced LD mapping), and **Fig. R1C4** (thickness, CD, LD' mapping as the radial direction).

The detailed changes are included in the changes in response to Reviewer #1's comment #3, Reviewer #1's comment #4, and Reviewer #2's comment #3.

Reviewer #4's comment #4: In supplementary figure 5, averaged polarization properties from Mueller matrices measured at rotation angles between 0° and 360° are reported. It is noticed that values of $\pm 45^\circ$ linear diattenuation (LD') and $\pm 45^\circ$ linear birefringence (LB') are of the same order than CD and circular birefringence (CB). It should be discussed the physical meaning of these results.

Response to comment #4: First, LD, LD', LB, and LB' describe matter's responses to linearly polarized inputs, while CD and CB describe responses to circularly polarized inputs. In this sense, they describe *different* physical phenomena. Hence, we do not think that the results and values from these two categories of optical responses can be compared directly. For this specific sample, the scale of LD, LD', LB, and LB' have peak values on the order of 0.2 - 0.3. The averages of LD, LD', LB, and LB' when samples are rotated one round (0° to 360° azimuthal angles) are all expected to be canceled out. **Figures R4C4(a)** and **R4C4(b)** display the LD' and LB' extracted from ellipsometry measurements at four in-

plane azimuthal rotation angles. LD' and LB' flip signs as the sample rotates. However, some experimental imperfections can lead to imperfect canceling. For example, the rotation axis of the rotation stage could be a bit off the center of the incident beam, so that different parts of the sample can be measured as the sample rotates. Indeed, after averaging, the peak residues of LD' and LB' are quite small (<0.02, 1 order of magnitude smaller) compared to their scales at each azimuthal rotation angle.

On the other hand, CD and CB have peak values on the order of 0.01 – 0.02. Also due to experimental imperfections, CD and CB display some dependence on the sample in-plane azimuthal rotation angle; see **Fig. R4C4(c)** and **Fig. R4C4(d)**. CD and CB spectra stay with the same spectral shape during rotation. There is no sign flip, which is in stark contrast with LD' and LB'. We took the average of CD and CB spectra as well for one round (0° to 360°) rotation to minimize the effects of such imperfections. The obtained averaged peak signals of CD and CB spectra are indeed close to their scales at each azimuthal rotation angle. From the obtained LD' spectra in **Fig R4C4(a)**, the CNT alignment direction is close to the line of $y = x$.

Fig. R4C4: (a) LD' spectra, (b) LB' spectra, (c) CD spectra, and (d) CB spectra at different in-plane sample rotation angles. (e) An illustration of CNT optical axis orientation.

In the revised manuscript, we have a few sentences in the Methods section. In the revised SI, we have added a new figure. The detailed changes are as follows:

Changes Made:

Main text, Line 633 - 638

Added: At each in-plane polarization angle, the amplitudes of LD, LD', LB, and LB' terms were on the order of 10^{-1} while the amplitudes of CD and CB terms were on the order of 10^{-2} . During the in-plane sample rotation, LD, LD', LB, and LB' terms flipped signs and nearly canceled out after averaging over spectra at all angles, while the CD and CB terms stayed with the same spectral shape (Supplementary Fig. 18).

SI

Added: Supplementary Figure 18.

Reviewer #4's comment #5: Regarding the twist-stacking method ii), appreciable values of linear diattenuation and linear birefringence are expected. The differential decomposition of the Mueller matrix of two-layer and three-layer stacks with a twist angle of 30° should be reported to demonstrate the equivalence with CD spectra shown in figure 4b.

Response to comment #5: We have performed new experiments of comparing CD signals from ellipsometry measurements and the standard CD spectrometer for the samples fabricated using twist-stacking. They also displayed good agreement. The purpose of comparing ellipsometry measurements with “four-configuration” measurements in a standard CD spectrometer is to validate the reported CD signals from “four-configuration” measurements are truly from structure-induced chirality, since ellipsometry measurements can separate the effects due to optical linear anisotropy from CD.

In the revised manuscript, we have a few sentences. In the revised SI, we have added new figures. The detailed changes are as follows:

Changes Made:

Main text, Line 226 - 227

Added red texts: To further confirm the validity of this four-configuration measurement approach, we performed spectroscopic ellipsometry measurements...

Main text, Line 265

Added red texts: Supplementary Figure 5 shows excellent agreement between the spectrum obtained through the four-configuration CD measurements and that from the ellipsometry measurements.

SI

Added: Supplementary Figures 4 and 5.

REVIEWER COMMENTS

Reviewer #1 (Remarks to the Author):

The authors have addressed almost all my questions satisfactorily, and I very much appreciate the additional experiments and analyses supplemented. I would like to recommend the manuscript for publication in Nature Communications after the following minor points:

1. The authors claimed that “the effective structures created by both mechanical rotation and twist-stacking methods are essentially the same, i.e., twisted stacks”. While the CD and LDr mappings are comprehensive, there is no actual data (e.g., microscopy images) supporting the claim. Some questions remain.

(a) If the authors zoom in on the calculated CD spectra in Fig R1C3-1(c), it is very likely that 2D spirals also produce nonzero, albeit small, chiroptical signals in the DUV. This is typical for 2D chirality. The authors should note this.

(b) The thickness-normalized ellipticity values of the twist-stacked assemblies are significantly larger than those obtained for assemblies fabricated by mechanical rotation. The CD values obtained experimentally for the latter are likely on the same order as per Fig R1C3-1(c). Wouldn't this hint at the reduced contribution of “twisted stacking” and more of a “spiral effect”? How are the two contributions decoupled for the mechanically rotated samples? Could the authors please comment on this?

2. I would like to simply point out that the rotating samples during CD spectral measurements is a common practice in the field. See for example Lv et al. *Angew. Chem. Int. Ed.* 56, 5055–5060 (2017). The added claim on Page 6 Line 268–270 should be removed or reworded accordingly.

Reviewer #2 (Remarks to the Author):

I thank the authors for their detailed answers to my questions and for the additions and corrections made to the manuscript. I think that with the new experiments performed, we have a much better understanding of the optical properties of the films (especially those obtained with the orbital shaker) and how they relate to the structure of the film. All my concerns and questions have been addressed, and I therefore recommend publication of the manuscript without further changes.

Reviewer #4 (Remarks to the Author):

In the revised manuscript the authors have attended all the comments previously arose. I recommend publication.

We thank all referees for their careful review of our manuscript and thoughtful comments. Below we address each of the questions/comments in detail:

Response to Reviewer #1

Reviewer #1's comment #1: The authors claimed that “the effective structures created by both mechanical rotation and twist-stacking methods are essentially the same, i.e., twisted stacks”. While the CD and LDr mappings are comprehensive, there is no actual data (e.g., microscopy images) supporting the claim. Some questions remain.

(a) If the authors zoom in on the calculated CD spectra in Fig R1C3-1(c), it is very likely that 2D spirals also produce nonzero, albeit small, chiroptical signals in the DUV. This is typical for 2D chirality. The authors should note this.

(b) The thickness-normalized ellipticity values of the twist-stacked assemblies are significantly larger than those obtained for assemblies fabricated by mechanical rotation. The CD values obtained experimentally for the latter are likely on the same order as per Fig R1C3-1(c). Wouldn't this hint at the reduced contribution of “twisted stacking” and more of a “spiral effect”? How are the two contributions decoupled for the mechanically rotated samples? Could the authors please comment on this?

Response to comment #1: We thank the reviewer for these valuable comments. However, we respectfully disagree that our claim is not supported by actual data. Instead, we believe that both our simulation and experimental data strongly support our claim. Please see our response to each of the sub-questions, (a) and (b), below:

Response to question (a): As suggested by the reviewer, we have zoomed in on the calculated CD spectra in Fig. R1C3-1(c) of the previous response letter and found that there is a very small CD signal in the 2D spiral structure. The calculated peak CD signal of the 2D spiral structure is ~ 0.4 mdeg/nm, which is **more than 2 orders of magnitude smaller (< 1%)** than the peak CD signal of the 3D helical structure (> 100 mdeg/nm). Hence, we conclude that the chiroptical response of the 2D spiral structure, if any, is **negligibly small** compared with that of the 3D helical structure.

Response to question (b): Our current understanding of the assembly formation process during mechanical-rotation-assisted vacuum filtration is as follows: carbon nanotubes (CNTs) are first deposited on the filter membrane to form layers of aligned structures, and then the alignment orientation starts rotating between layers by the flow field created by mechanical rotation, which lasts for a short period of time (e.g., 30 seconds). When the mechanical rotation speed is in the intermediate range (e.g., 140 RPM), the created flow field produces a spiral pattern of alignment orientation on top of macroscopic alignment layers (see Supplementary Figures 10 and 14). Hence, a 3D helical twisted stacking structure and a 2D spiral structure coexist in samples fabricated through mechanical rotation at 140 RPM, and both can in principle contribute to the chiroptical response. Therefore, the 2D spiral pattern is a part of the 3D twisted stacking structure (see Supplementary Figure 14), and thus, they are not decoupled. *What we claim* is that the experimentally observed CD signals in samples fabricated through mechanical rotation are **mainly** from the formed 3D helical structure, which is the same as samples fabricated using the twist-stacking approach. This claim is supported by the following pieces of evidence from actual data:

(1) Based on the simulation data mentioned in our response to Question (a), the chiroptical response of the 2D spiral structure, if any, is negligibly small (<1%) compared with that of the 3D helical structure.

(2) Experimentally, such a small CD signal coming from a 2D spiral structure (e.g., < 0.5 mdeg/nm) is unlikely to be captured in our CD “4-configuration” measurement. As mentioned in Line 222 of the

main text and Supplementary Figure 3b, an aligned CNT film with a racemic mixture, which is expected to have zero CD, displays a residual CD signal < 0.5 mdeg/nm because of imperfect cancellation in the “4-configuration” measurement. Hence, the small CD signal that comes from a 2D spiral structure is within or below the minimum CD signal we can deterministically measure in experiments.

(3) Furthermore, as pointed out by the reviewer, the thickness-normalized ellipticity values of the twist-stacked assemblies, which do not have any 2D spiral structures, are generally larger than those obtained for assemblies fabricated by mechanical rotation. However, their values are both much larger than what can be observed in 2D spiral structures. In some cases, the samples fabricated through twist stacking and mechanical rotation show quite close ellipticity values. For example, the 2-layer twist-stacked sample with a rotation angle of 15° shown in Fig. 2a of the main text has a thickness-normalized ellipticity value of ~ 7 mdeg/nm. This value is quite close to the value in the sample fabricated through clockwise mechanical rotation, which is ~ 6 mdeg/nm, as shown in Fig. 4b of the main text. In addition, for mechanical-rotation-created samples, the rotation angles are difficult to optimize; the mechanical rotation only lasts for 30 seconds, so that only a small portion of produced films contains 2D spiral structures to form 3D helical structures. Both factors can lead to smaller ellipticity values for mechanical rotation samples compared to twist-stacked samples. Hence, we do not believe that the larger ellipticity values observed in twist-stacked samples compared to mechanical-rotation-created samples imply more contributions from 2D spiral structures.

(4) Moreover, the experimentally observed distribution of CD signs in the CD mapping of mechanical-rotation-fabricated samples, shown in Supplementary Figure 13, provides another strong piece of evidence to support our claim. If the chiroptical response contribution from the 2D spiral structure were a dominating factor, a single handedness of the spiral structure (e.g., clockwise) would lead to the **same** CD signs across the sample, which directly contradicts the observation that there are two types of handedness coexisting in the same sample. In contrast, not only the coexistence of two types of handedness but also the distribution of CD signs can be explained well by assuming twisted stacking between the bottom aligned layers and the formed 2D spiral structures on top; see Supplementary Figure 14 and the corresponding descriptions in the main text.

In summary, we believe that our current simulation and experimental results strongly support our claim. Finally, we would like to have a few remarks on microscopy imaging. In order to observe the direction of alignment in CNT samples, high-resolution scanning electron microscopy is typically used. Then, it is needed to perform precise and clean layer-by-layer removal of aligned CNT layers to observe any alignment orientation change along the film thickness direction. However, as mentioned in our previous response letter, such precise and clean layer-by-layer removal is challenging and not a trivial task. We have tried a reactive dry etching approach, but the results were not satisfactory. A specially designed process, such as one developed for graphene (<https://www.science.org/doi/10.1126/science.1199183>), would be needed. The design of such a process is beyond the scope of this manuscript but is worth future investigations.

In the revised manuscript, we have added a new sentence for further clarification. The detailed changes are as follows:

Changes Made:

Main text, Line 404 – 410

Added: The chiroptical response contribution from a 2D spiral structure is negligibly small compared with that from a 3D helical structure. Furthermore, the fact that both chiralities were observed in a sample with only one rotation handedness and the CD sign spatial distribution within the sample shown in Supplementary Fig. 13 also confirm that the observed chiroptical response mainly originates from the 3D helical structure.

Reviewer #1's comment #2: I would like to simply point out that the rotating samples during CD spectral measurements is a common practice in the field. See for example Lv et al. *Angew. Chem. Int. Ed.* 56, 5055–5060 (2017). The added claim on Page 6 Line 268–270 should be removed or reworded accordingly.

Response to comment #2: The reviewer is correct, and we thank him/her for making us aware of this reference. In light of this comment, **in the revised manuscript, we have removed the claim from Lines 268 – 270 and have added the mentioned reference.**